# EXTENDED INDUCTIVE REASONING FOR PERSONAL-IZED PREFERENCE INFERENCE FROM BEHAVIORAL SIGNALS

## ABSTRACT

While large language models (LLMs) excel at deductive reasoning tasks such as math and coding, their capacity for inductive reasoning, which involves deriving general rules from incomplete evidence, remains underexplored. This paper investigates extended inductive reasoning in LLMs through the lens of personalized preference inference, a critical challenge in LLM alignment where current approaches struggle to capture diverse user preferences. The task demands strong inductive reasoning capabilities as user preferences are typically embedded implicitly across various interaction forms, requiring models to synthesize consistent preference patterns from scattered signals. We propose ALIGNXPLORE, a model that leverages extended reasoning chains to enable systematic preference inference from behavioral signals in users' interaction histories. Such explicit preference articulation enables efficient streaming inference: when new behavioral signals emerge, the model can directly build upon previously inferred preference descriptions rather than reprocessing historical signals from scratch, while also supporting iterative refinement to the inferred preferences. We develop ALIGNXPLORE by combining cold-start training based on synthetic data with subsequent online reinforcement learning. Extensive experiments demonstrate that ALIGNXPLORE achieves substantial improvements over the backbone model by an average of 15.49% on both in-domain and out-of-domain benchmarks, while maintaining a strong generalization ability across different input formats and downstream models. Further analyses establish best practices for preference inference learning through systematic comparison of reward modeling strategies, while revealing the emergence of human-like inductive reasoning patterns during training.

## 1 INTRODUCTION

Large language models (LLMs) have achieved great success in complex, deductive-heavy tasks such as code generation (Chen et al., 2021) and mathematical problem-solving (Lightman et al., 2023) by applying known rules through extended reasoning chains (DeepSeek-AI, 2025; Morsanyi et al., 2018; Chollet, 2019). In contrast, inductive reasoning, i.e., the ability to derive general rules from incomplete and noisy evidence (Hayes et al., 2010), presents a significant yet underexplored challenge. Despite being a cornerstone of human intelligence and scientific discovery (Heit, 2000; Ferrara et al., 1986; Kinshuk et al., 2006), extending LLMs' reasoning abilities to complex inductive tasks has received limited attention.

We investigate extended inductive reasoning through the lens of personalized preference inference: the task of synthesizing an explicit summary of a user's preferences from a scattered history of their implicit behavioral signals. This investigation is critical for two reasons. First, it addresses a key limitation in LLM alignment. Current methods targeting universal values like helpfulness and harmlessness (Askell et al., 2021; Ouyang et al., 2022; Bai et al., 2022; Achiam et al., 2023; Team, 2024) struggle with diverse individual preferences (Kirk et al., 2023), leading to reduced user satisfaction and potential biases (Siththaranjan et al., 2024; Guan et al., 2025; Tong, 2023). Second, preference inference serves as a prime testbed for complex induction. Unlike deductive tasks, there are no pre-defined rules; preferences are rarely stated explicitly (Lee et al., 2024) and must be inferred from heterogeneous signals like textual posts (Wu et al., 2025b), behavioral choices (Ouyang et al.,

2022), and demographics (Zhang et al., 2018). The core challenge is thus to identify underlying patterns within this noisy data and generalize them to new contexts, as illustrated in Figure 1.

Despite the critical importance, existing personalization methods often bypass explicit preference inference, instead using direct mappings that incorporate implicit signals as prompts (Xu et al., 2022; Lee et al., 2024), parameters (Kang et al., 2023; Tan et al., 2024), or hidden representations (Poddar et al., 2024; Ning et al., 2024). This approach renders the preference modeling process opaque and, more critically, inefficient: these methods cannot incrementally refine preferences when new behavioral signals become available, forcing models to process growing interaction histories from scratch, creating a significant scalability bottleneck. To address these issues, we propose ALIGNXPLORE, a model that leverages extended[1] reasoning chains for systematic preference inference. By framing personalization as the task of articulating an explicit preference summary, ALIGNXPLORE unlocks a powerful and efficient streaming inference mechanism. The inferred preference acts as a compact, updatable state, allowing the model to build upon prior inferences when new signals arrive, rather than reprocessing historical data from scratch. We train our model using a two-stage framework: first, we leverage synthetic data for supervised fine-tuning (SFT) to address the cold-start problem, and then we use reinforcement learning (RL) to optimize the model's reasoning capabilities for accurate preference inference.

Extensive experiments show ALIGNXPLORE substantially improves personalized alignment on both in-domain and out-of-domain benchmarks, outperforming its backbone by 15.49% and competing with much larger models like GPT-4 (Achiam et al., 2023) and DeepSeek-R1-671B (DeepSeek-AI, 2025). Its streaming mechanism is not only efficient, avoiding recomputation over growing histories, but also boosts performance via gradual preference refinement. We attribute the model's strong generalization and robustness to its extended reasoning, which fosters systematic inductive patterns over superficial correlations. Further analysis reveals two key findings: (1) optimizing for preference judging yields more stable training than for response generation, suggesting a best practice; and (2) our two-stage training mirrors human induction (Heit, 2000), where cold-start training establishes basic characterization and RL refines it into actionable hypotheses.

The main contributions of this work are as follows: (1) We are the first to systematically investigate extended inductive reasoning in LLMs via personalized preference inference, showing how structured reasoning can derive generalizable preferences from implicit signals. (2) We develop ALIGNXPLORE, a preference inference model featuring an efficient streaming mechanism. It is trained using a novel two-stage framework that combines synthetic data with reinforcement learning. (3) We demonstrate through comprehensive evaluations that ALIGNXPLORE achieves substantial improvements over existing methods in performance, efficiency, generalization, and robustness. Our analyses also offer insights into reward modeling and the emergence of inductive reasoning[2].

## 2 RELATED WORKS

**Inductive reasoning** Inductive reasoning, i.e., making probabilistic generalizations from incomplete evidence (Lake et al., 2017; Hayes et al., 2010), is crucial for cognitive tasks like scientific discovery (Holland, 1986) and has gained renewed attention in LLM evaluation via benchmarks like the Abstract Reasoning Corpus (ARC) (Chollet, 2019; Moskvichev et al., 2023). While prior work on LLM induction often focuses on few-shot generalization (Radford et al., 2018; Wang et al., 2024b), we argue that preference inference presents a more complex and realistic testbed. It introduces distinct challenges, such as reasoning over unstructured language rather than formal systems (Qiu et al., 2024; Yan et al.), handling heterogeneous signals, and learning from negative examples (Laskin et al., 2023). Our framework addresses these challenges in a principled, interpretable way.

**Extended reasoning in LLMs** Recent advances in extended reasoning have moved beyond the shallow, linear steps of traditional Chain-of-Thought (Wei et al., 2022) to significantly boost LLM performance (OpenAI, 2024; Chen et al., 2025b). This progress is driven by three key mechanisms: (1) in-depth logical chains in diverse formats like natural language (Wang et al., 2024a), formal

---

[1]Here, we define "extended" not by algorithmic complexity, but by the deliberate, multi-step thought process a model undertakes before reaching a conclusion, which is a paradigm that has proven effective in complex deductive tasks (OpenAI, 2024; Claude, 2025)

[2]Code is available at `https://anonymous.4open.science/r/ICLR2026-AlignXplore`

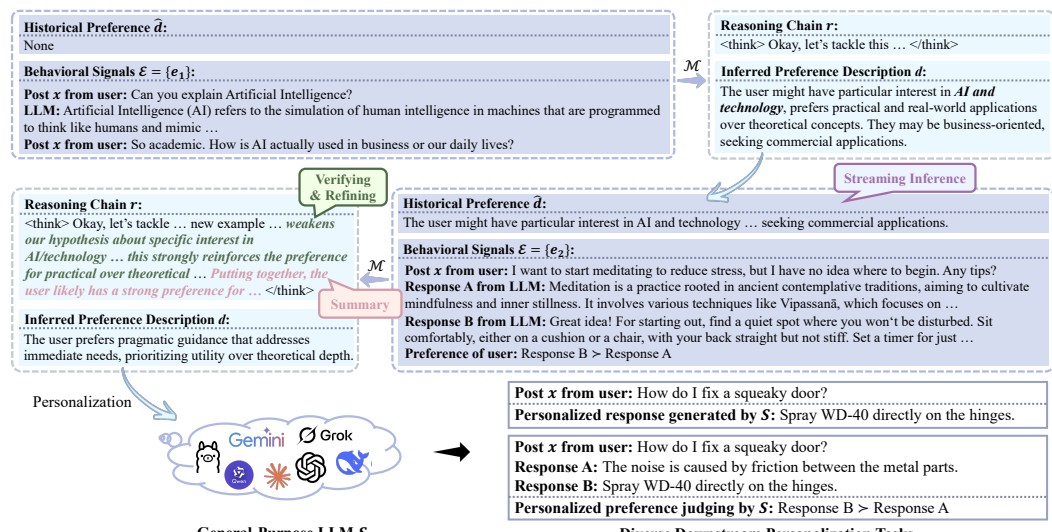

Figure 1: Preference inference task overview. Our model performs human-like inductive reasoning for preference inference by progressively refining its preference hypotheses through iterative testing and validation. These inferred preferences can then guide diverse downstream personalization tasks.

language (Wen et al., 2025), or latent space (Hao et al., 2024); (2) systematic exploration of solution spaces using techniques like RL-trained internal mechanisms (DeepSeek-AI, 2025) or external search algorithms (Zhang et al., 2024; Yao et al., 2024; Snell et al., 2025); and (3) iterative self-reflection for verification and correction, powered by SFT (Team, 2024; Gandhi et al., 2025) or RL with verifiable rewards (DeepSeek-AI, 2025; Yu et al., 2025). While this paradigm has proven effective in deductive tasks such as math (Hu et al., 2025a), coding (Jain et al.), scientific question-answering (Rein et al., 2024), reward modeling (Chen et al., 2025c), and multimodal reasoning (Wu et al., 2025a), we are the first to extend it to preference inference, a domain demanding strong inductive reasoning.

**Personalized alignment** Motivated by the limitations of one-size-fits-all alignment (Askell et al., 2021; Kirk et al., 2023; Siththaranjan et al., 2024), personalized alignment aims to adapt LLMs to individual preferences (Kalai & Smorodinsky, 1975; Oldenburg & Zhi-Xuan, 2024). Key challenges in this area include: (1) **Preference inference** from scattered, implicit signals (Wu et al., 2025b; Ouyang et al., 2022; Zhang et al., 2018; Guan et al., 2025), where current works often retrieve relevant contexts rather than perform explicit inference, limiting their accuracy (Zhao et al., 2025; Pan et al., 2025; Zhang, 2023; Li et al., 2025). (2) **Preference modeling** via direct prompts (Xu et al., 2022; Lee et al., 2024), parameters (Kang et al., 2023; Tan et al., 2024), or latent states (Poddar et al., 2024; Ning et al., 2024); our work focuses on interpretable, model-agnostic prompting. (3) **Feedback-driven alignment**, which can occur during training (Jang et al., 2023; Guan et al., 2024; Kuang et al., 2024) or at inference time (Shi et al., 2024; Chen et al., 2025a; Rame et al., 2023). Our work is the first to integrate extended reasoning for accurate preference inference with efficient mechanisms for handling evolving user interactions (Chandrashekaran et al., 1996).

## 3 METHODOLOGY

This section first formulates the preference inference task and its evaluation (§3.1), then details the two-stage training strategy: a cold-start phase to build basic reasoning capabilities (§3.3), followed by an RL phase to directly optimize for reward (§3.4). Figure 1 presents our task overview, and Figure 2 illustrates the entire training recipe.

### 3.1 TASK FORMULATION

We formulate the personalized preference inference task as follows: given a set of a user's behavioral signals $\mathcal{E} = \{e_1, \ldots, e_T\}$ and an optional historical preference summary $\hat{d}$, the model $\mathcal{M}$ must generate an explicit preference description $d$ in natural language, supported by an extended reasoning

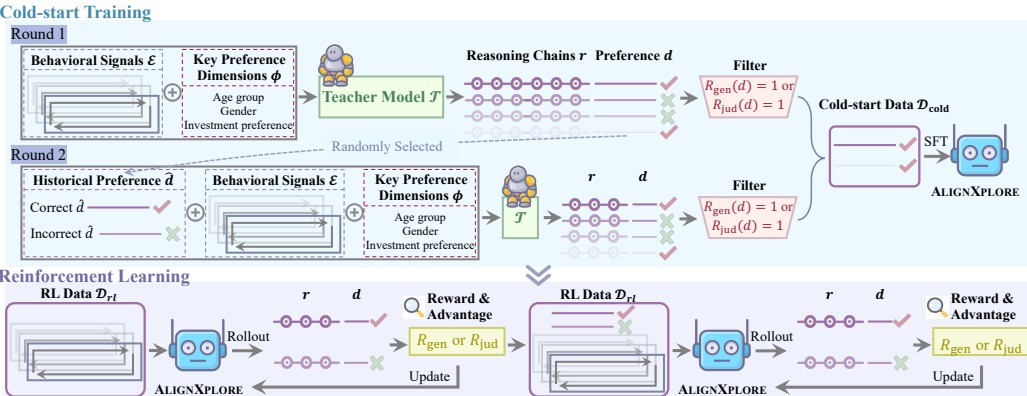

Figure 2: Two-stage training process of ALIGNXPLORE, which combines cold-start training using synthetic data from teacher models with reinforcement learning optimization to enhance the model's reasoning capabilities.

chain $r$. Formally:

$$(r, d) = \mathcal{M}(\mathcal{E}, \hat{d}), \tag{1}$$

where $\hat{d}$ is a summary of past preferences, which is left empty for the initial inference. The behavioral signals $\mathcal{E}$ can be diverse; our main experiments use comparative judgments (a prompt with preferred/less-preferred responses), but the method is format-agnostic and can accommodate other signals like user posts or reviews (see §4.3). The output preference description $d$ is designed to be model-agnostic, enabling it to condition any general-purpose LLM $\mathcal{S}$ for downstream personalization tasks (Lee et al., 2024; Li et al., 2025).

This formulation is crucial for real-world applications as it enables an efficient *streaming inference mechanism*. The historical preference description $\hat{d}$ serves as a compact, stateful summary of all past interactions. Consequently, when new behavioral signals arrive, the model can update its preference incrementally by processing only the new signals and this summary, without accessing the full, raw history. This approach fundamentally overcomes a major scalability bottleneck of prior methods based on direct prompting (Xu et al., 2022; Li et al., 2025) and parameter updates (Kang et al., 2023; Poddar et al., 2024), which lack such an explicit state and are therefore forced to reprocess the entire, ever-growing user history for each update.

## 3.2 EVALUATION FRAMEWORK

We assess the quality of a generated preference description $d$ by its ability to guide a downstream model $\mathcal{S}$ to make choices aligned with the user's true preferences. While a real-time user reward would be ideal, it is impractical for large-scale evaluation. We, therefore, adopt an efficient offline evaluation using held-out comparative judgment data. Given a user post $x$ with a known preferred response $y_w$ and a non-preferred one $y_l$, we define our offline reward as:

$$R_{\text{offline}} = \mathbb{1}\big(f(y_w; x, d, \psi_f) > f(y_l; x, d, \psi_f)\big)R_{\text{format}}, \tag{2}$$

where $f$ is a scoring function that quantifies how well a response aligns with the user's inferred preference, the term $\psi_f$ represents all other necessary inputs for scoring, and $R_{\text{format}} \in \{0, 1\}$ validates whether the model's output structure satisfies the requirement (Appendix C). The scoring function $f$ can be instantiated in several ways depending on the downstream model $\mathcal{S}$.

First, when $\mathcal{S}$ is a generation model (denoted as $\mathcal{S}_{\text{gen}}$) (Rafailov et al., 2024), the score is the log-probability gain of the response when conditioned on the preference description $d$. Here, the scoring function $f$ is instantiated as $f(y_{w/l}; x, d) = \log(\frac{\mathcal{S}_{\text{gen}}(y_{w/l}|x,d)}{\mathcal{S}_{\text{gen}}(y_{w/l}|x)})$, so $\psi_f$ corresponds to $\emptyset$. This yields the reward $R_{\text{gen}}$:

$$R_{\text{gen}} = \mathbb{1}\big(\log \frac{\mathcal{S}_{\text{gen}}(y_w|x, d)}{\mathcal{S}_{\text{gen}}(y_w|x)} > \log \frac{\mathcal{S}_{\text{gen}}(y_l|x, d)}{\mathcal{S}_{\text{gen}}(y_l|x)}\big)R_{\text{format}}. \tag{3}$$

Alternatively, when $\mathcal{S}$ is a preference judging model (denoted as $\mathcal{S}_{\text{jud}}$) (Zheng et al., 2023), it directly outputs the probability of a response being preferred. In this case, the scoring function is the model's

direct output, $f(y_{w/l}; x, d, y_w, y_l) = \mathcal{S}_{\text{jud}}(y_{w/l}|x, d, y_w, y_l)$, thus $\psi_f = \{y_w, y_l\}$. The reward $R_{\text{jud}}$ is then:

$$R_{\text{jud}} = \mathbb{1}\big(\mathcal{S}_{\text{jud}}(y_w|x, d, y_w, y_l) > \mathcal{S}_{\text{jud}}(y_l|x, d, y_w, y_l)\big) R_{\text{format}}. \tag{4}$$

While other instantiations exist (Meng et al., 2024), we primarily use $R_{\text{jud}}$ for training and evaluation due to its stability, analyzing $R_{\text{gen}}$ in our ablations.

## 3.3 COLD-START TRAINING

To address the cold-start problem where small models struggle with complex inference from instructions alone, we construct a high-quality synthetic dataset via two stages. For each set of preference signals $\mathcal{E}$, for each example in the original preference signals $e_i \in \mathcal{E}$, we first identify key preference dimensions $\phi$ expressed in natural language that potentially reveal user preferences, which serve as analytical guidance for subsequent preference inference. We then prompt an advanced teacher model $\mathcal{T}$ with both these identified dimensions $\phi$ and the original implicit signals $\mathcal{E}$ to generate $G$ reasoning chains and preference descriptions (see Appendix C for prompt templates), as exemplified in Figure 2.

To support streaming inference, we further prompt the teacher model $\mathcal{T}$ with a new set of behavioral signals of the same user and a prior preference description $\hat{d}$ which is randomly selected from $\{d_i\}_{i=1}^G$. We then mix these new examples with the original ones and apply outcome-based verification, retaining only those that achieve an optimal reward score. Finally, we train $\mathcal{M}$ parameterized by $\theta$ by maximizing the log-likelihood of the target sequences:

$$\mathcal{L}_{\text{cold}} = -\mathbb{E}_{(r,d)\sim\mathcal{T}} \frac{1}{|r| + |d|} \sum_t \log p_\theta(\{r, d\}_t | \mathcal{E}, \hat{d}), \tag{5}$$

where $\{r, d\}_t$ is the $t$-th token and $\hat{d}$ may be empty for first-turn generation. Appendix B.8 further discusses the influence of the quality and diversity of the cold-start data on model performance.

## 3.4 REINFORCEMENT LEARNING

We further refine the model's reasoning capabilities beyond the cold-start phase using RL. We adopt Group Relative Policy Optimization (GRPO) (DeepSeek-AI, 2025), an algorithm effective for long-horizon reasoning. Following Hu et al. (2025a), we remove the KL penalty for more stable optimization. To support streaming inference, the rollout process mirrors our cold-start data generation by involving two rounds for each user: (1) We generate a set of $G$ initial outputs $\{(r_j, d_j)\}_{j=1}^G \sim p_{\text{old}}(\cdot|\mathcal{E})$, where $p_{\text{old}}$ is the old policy model. (2) We generate a second set of $G$ outputs $\{(r_k, d_k)\}_{k=1}^G \sim p_{\text{old}}(\cdot|\mathcal{E}', \hat{d})$, where $\mathcal{E}'$ is a distinct set of signals from the same user and $\hat{d}$ is a historical preference sampled from the output of the first round. The GRPO objective is then optimized over the union of outputs from both rounds:

$$\mathcal{L}_{\text{RL}} = -\frac{1}{2G} \sum_{i=1}^{2G} \frac{1}{|r_i| + |d_i|} \rho_i, \tag{6}$$

$$\rho_i = \sum_t \min\big(\frac{p_\theta(\{r_i, d_i\}_t | \mathcal{E}, \hat{d})}{p_{\text{old}}(\{r_i, d_i\}_t | \mathcal{E}, \hat{d})} A_i, \text{clip}(\frac{p_\theta(\{r_i, d_i\}_t | \mathcal{E}, \hat{d})}{p_{\text{old}}(\{r_i, d_i\}_t | \mathcal{E}, \hat{d})}, 1 - \epsilon, 1 + \epsilon) A_i\big), \tag{7}$$

$$A_i = \frac{R_i - \text{mean}(\{R_j\}_{j=1}^G)}{\text{std}(\{R_j\}_{j=1}^G)}, \tag{8}$$

where $\hat{d}$ is empty for the rollout of the first round, and $\rho_i$ is computed based on the corresponding advantage $A_i$ normalized within each respective round. The rewards $R_i$ are from Eq. 3 or 4. While this two-round process can be naturally extended to multiple rounds, we found it strikes an effective balance between performance gains and computational cost.

Table 1: Offline preference inference accuracy ($\text{ACC}_{\text{jud}}$, %) with Qwen2.5-7B-Instruct as the preference judging model. "Extended Reasoning": whether generating long reasoning chains. "#Signals": the number of pairs in $\mathcal{E}$; $\hat{d}$: if a historical preference summary is used. Pink-shaded rows show results with 4 additional pairs ($T = 8$ or $T = 4, \hat{d} \neq \emptyset$). Gray-shaded rows are large-sized models or golden preference baselines, where *italicized* scores are lower than the best result (**bold**) among smaller models. Underlined is second-best. * indicates statistical significance ($p < 0.05$, $t$-test).

| Method | Extended Reasoning | Setting | #Signals | $\hat{d}$ | $\text{ALIGNX}_{\text{test}}$ | P-SOUPS | | |
|---|---|---|---|---|---|---|---|---|
| | | | | | | Informativeness | Style | Expertise |
| *Directly given preference descriptions* | | | | | | | | |
| Null | N/A | N/A | N/A | N/A | 51.37* | 45.85* | 17.00* | 36.00* |
| $\mathcal{E}$ | N/A | N/A | 4 | N/A | 50.33* | 41.03* | 37.33* | 36.00* |
| **Golden Preference** | N/A | N/A | N/A | N/A | *64.63* | 68.94 | 84.50 | 90.17 |
| *Previous specialized methods for inductive reasoning and personalization* | | | | | | | | |
| **LMInductReason** (Qiu et al., 2024) | N/A | Base | 4 | ✗ | 51.80* | 44.35* | 27.50* | 38.17* |
| **VPL** (Poddar et al., 2024) | N/A | Base | 4 | ✗ | 50.73* | 44.19* | 52.50* | 51.00* |
| **EXPO** (Hu et al., 2025b) | N/A | Base | 4 | ✗ | 44.30* | 51.50* | 76.17* | 55.17* |
| **PBA** (Li et al., 2025) | N/A | Base | 4 | ✗ | 62.77* | 53.65* | 31.33* | 50.50* |
| *Preference descriptions generated by state-of-the-art LLMs* | | | | | | | | |
| **Qwen2.5-7B-Instruct** (Team, 2024) | ✗ | Base | 4 | ✗ | 56.33* | 53.82* | 59.00* | 65.17* |
| **DS-R1-Distill-Qwen-7B** (DeepSeek-AI, 2025) | ✓ | Base | 4 | ✗ | 57.63* | 51.16* | 45.83* | 56.67* |
| **DS-R1-Distill-Qwen-7B** (DeepSeek-AI, 2025) | ✓ | Base | 8 | ✗ | 56.13* | 49.50* | 56.17* | 57.33* |
| **DS-R1-Distill-Qwen-7B** (DeepSeek-AI, 2025) | ✓ | Streaming | 4 | ✓ | 56.40* | 50.00* | 49.50* | 60.17* |
| **Qwen3-32B**$_{\text{non-thinking}}$ (Yang et al., 2025) | ✗ | Base | 4 | ✗ | *57.60* | *54.98* | *61.50* | *66.67* |
| **GPT-4** (Achiam et al., 2023) | ✗ | Base | 4 | ✗ | *66.10* | *53.82* | *73.33* | *71.83* |
| **QwQ-32B** (Team, 2025) | ✓ | Base | 4 | ✗ | *65.70* | *58.14* | *72.17* | *71.50* |
| **Qwen3-32B**$_{\text{thinking}}$ (Yang et al., 2025) | ✓ | Base | 4 | ✗ | *65.03* | *57.14* | *71.67* | *73.83* |
| **DeepSeek-R1-671B** (DeepSeek-AI, 2025) | ✓ | Base | 4 | ✗ | *70.47* | *55.48* | *79.66* | *76.17* |
| **DeepSeek-R1-671B** (DeepSeek-AI, 2025) | ✓ | Base | 8 | ✗ | *70.23* | *56.98* | *84.17* | *79.17* |
| **DeepSeek-R1-671B** (DeepSeek-AI, 2025) | ✓ | Streaming | 4 | ✓ | *67.70* | *56.64* | *69.50* | *69.17* |
| *Preference descriptions generated by our preference inference model* | | | | | | | | |
| **ALIGNXPLORE-7B** | ✓ | Base | 4 | ✗ | 65.33 | 54.32 | 69.67 | 63.83 |
| **ALIGNXPLORE-7B** | ✓ | Base | 8 | ✗ | 64.30 | 57.14 | 70.33 | 66.50 |
| **ALIGNXPLORE-7B** | ✓ | Streaming | 4 | ✓ | **71.47** | **61.30** | **83.00** | **71.33** |
| **ALIGNXPLORE-7B w/o RL** | ✓ | Base | 4 | ✗ | 61.80 | 52.82 | 54.00 | 59.83 |
| **ALIGNXPLORE-7B w/o Cold-start** | ✓ | Base | 4 | ✗ | 62.80 | 56.64 | 64.83 | 59.50 |

# 4 EXPERIMENTS

## 4.1 EXPERIMENTAL SETUP

**Implementation details** We simulate a two-round streaming setting for both cold-start data generation and RL, where the first round uses 4 behavioral signals (i.e. $T = 4$ in $\mathcal{E}$) and the second uses another 4 signals plus the inferred preference from the first round. This process is naturally extensible to more rounds. We use DeepSeek-R1-Distill-Qwen-7B (DeepSeek-AI, 2025) as our backbone and explore other backbones in Appendix B.6. We train the model on the ALIGNX dataset (Li et al., 2025), a comprehensive personalized alignment dataset spanning 90 preference dimensions with balanced positive and negative examples. We create two 7,000-instance sets for cold-start and RL training, respectively. For cold-start data generation, QwQ-32B (Team, 2025) serves as the teacher model. We use the $R_{\text{jud}}$ reward (Eq. 4) with Qwen2.5-7B-Instruct as the preference judging model for both cold start and RL, with a batch size of 128 and $G = 4$ rollouts per prompt. For inference, we use nucleus sampling ($p = 0.95$, $k = 10$) with a temperature of 0.9 (Holtzman et al., 2020; Fan et al., 2018; Goodfellow et al., 2014). To analyze different configurations, we also train two non-streaming baseline models with 4 and 8 examples in $\mathcal{E}$, respectively. Appendix B.1 shows more details.

**Benchmarks** We evaluate on two benchmarks: (1) $\text{ALIGNX}_{\text{test}}$ (Li et al., 2025), the official test set of ALIGNX; and (2) P-SOUPS (Jang et al., 2023), an out-of-domain benchmark which focuses on three preference dimensions: "expertise," "informativeness," and "style." Table 6 in Appendix B.2 summarizes the statistics. Following our training settings, we consider two evaluation settings: a base setting where models perform inference using 4 or 8 preference pairs with empty $\hat{d}$, and a streaming setting where the model first uses the initial 4 pairs to infer a historical preference description $\hat{d}$, then combine $\hat{d}$ with 4 new pairs to generate the final preference description. All preference pairs are

Table 2: Online preference inference evaluation results (GPT-4 win rate, %, row model against column model) using Qwen2.5-7B-Instruct as the personalized response generation model. We randomly select 400 test cases per benchmark for evaluation. **M1**: Qwen2.5-7B-Instruct; **M2**: DS-R1-Distill-Qwen-7B; **M3**: ALIGNXPLORE-7B.

| ALIGNX$_{\text{test}}$ | M1 | M2 | M3 | P-SOUPS | M1 | M2 | M3 |
|:---:|:---:|:---:|:---:|:---:|:---:|:---:|:---:|
| M1 | - | 43.00 | 37.00 | M1 | - | 51.33 | 42.33 |
| M2 | 57.00 | - | 43.00 | M2 | 48.67 | - | 46.67 |
| M3 | 63.00 | 57.00 | - | M3 | 57.67 | 53.33 | - |

randomly sampled from the same user's behavioral signals. We ensure that each model is evaluated under its corresponding training setting.

**Evaluation metrics**  Due to the inherent difficulty of directly evaluating preference inference quality, we employ both offline and online metrics for indirect evaluation: **(1) Offline evaluation:** We compute the accuracy of preference-guided response generation (**ACC$_{\text{gen}}$**) and preference judging (**ACC$_{\text{jud}}$**). These metrics are calculated as the average of the binary rewards $R_{\text{gen}}$ (Eq. 3) and $R_{\text{jud}}$ (Eq. 4), respectively, over the entire test set. We primarily focus on **ACC$_{\text{jud}}$** as it aligns with our training objective. **(2) Online evaluation:** We introduce **GPT-4 Win Rate**,[3] where GPT-4 conditioned on the ground-truth preferences (provided by the benchmarks) compares responses generated given preference descriptions from different models (Kumar et al., 2024; Jang et al., 2023).[4]

**Baselines**  We compare our approach with three groups of baselines: **(1) Direct preference descriptions,** where different forms of preference information are directly inserted into the downstream model's prompt. This includes *Null*, which provides no preference context to establish a non-personalized baseline; $\mathcal{E}$ (raw behavioral signals), which concatenates the raw behavioral signals to condition the model; and *Golden Preference*, which uses the benchmark's ground-truth descriptions to simulate perfect inference, serving as a conceptual upper bound[5]). **(2) Specialized methods,** including *LMInductReason* (Qiu et al., 2024) for inductive reasoning, *VPL* (Poddar et al., 2024) and *EXPO* (Hu et al., 2025b) for preference modeling, and *PBA* (Li et al., 2025) for structured preference prediction. **(3) State-of-the-art LLMs,** including small models (*Qwen2.5-7B-Instruct* (Team, 2024), *DS-R1-Distill-Qwen-7B* (DeepSeek-AI, 2025)) and large models (*QwQ-32B* (Team, 2025), *Qwen3-32B* (Yang et al., 2025), *GPT-4* (Achiam et al., 2023), *DeepSeek-R1-671B*[6] (DeepSeek-AI, 2025)), which are prompted with the raw behavioral signals to generate a user preference description.

We also evaluate ablated versions of our model (w/o RL and w/o Cold-start) to verify the effectiveness of each training stage. See Appendix B.3 for baseline implementation details.

## 4.2 MAIN RESULTS

**Offline evaluation**  Table 1 presents our main offline evaluation results, leading to five key findings: **(1) Preference inference is necessary.** Directly using behavioral signals ($\mathcal{E}$) performs no better than the "Null" baseline and far worse than using golden preferences. **(2) Prior methods are limited.** Prompt-based (LMInductReason) and latent-variable (VPL, EXPO) methods perform poorly. Even PBA, which uses predefined preferences, generalizes poorly to out-of-domain data (P-SOUPS). **(3) Extended reasoning is superior.** Models with extended reasoning consistently outperform their counterparts with concise reasoning (e.g., Qwen3-32B$_{\text{thinking}}$ vs. Qwen3-32B$_{\text{non-thinking}}$: 65.03% vs. 57.60%; DeepSeek-R1-671B vs. GPT-4: 70.47% vs. 66.10%). **(4) ALIGNXPLORE is significantly superior.** Our model surpasses same-sized baselines and competes with much larger models like Qwen3-32B and GPT-4, even outperforming the golden preference baseline on ALIGNX$_{\text{test}}$. **(5) Our streaming mechanism is both efficient and effective.** While the backbone model gains little from extra signals, ALIGNXPLORE significantly benefits from its streaming mechanism, outperforming even its non-streaming 8-pair variant. This shows our model effectively utilizes historical information

---

[3]We use OpenAI's API "gpt-4-turbo-2024-04-09" for all our subsequent experiments.

[4]We present the training and inference costs in Appendix B.7.

[5]Note that golden preference descriptions, while semantically accurate, may not necessarily lead to optimal downstream personalization performance due to potential gaps in model compatibility.

[6]All experiments are based on the DeepSeek-R1 version released on 2025/01/20.

Table 3: Generalization evaluation ($ACC_{jud}$, %) in terms of two aspects: *(1) Input-form*: "ALIGNX$_{test}$ w/ UGC" column shows accuracy when inferring from user-generated content instead of preference pairs. *(2) Cross-model*: The remaining columns show the transferability of inferred preferences (rows) to different downstream models (columns) on the original ALIGNX$_{test}$ benchmark.

| Method | Extended Reasoning | ALIGNX$_{test}$ w/ UGC | Preference Judging Model $\mathcal{S}_{jud}$ | | |
| --- | --- | --- | --- | --- | --- |
| | | | Qwen2.5-7B-Instruct | QwQ-32B | DeepSeek-R1-671B |
| $\mathcal{E}$ | N/A | 52.17 | 50.33 | 49.03 | 50.12 |
| **Golden Preference** | N/A | 69.87 | *64.63* | 74.30 | 78.97 |
| **Qwen2.5-7B-Instruct** | ✗ | 57.57 | 56.33 | 56.90 | 58.15 |
| **DS-R1-Distill-Qwen-7B** | ✓ | 58.30 | 57.63 | 58.70 | 59.61 |
| **DeepSeek-R1-671B** | ✓ | *61.97* | 70.47 | 73.73 | 74.00 |
| **ALIGNXPLORE-7B** | ✓ | **61.97** | **65.33** | **68.53** | **67.59** |

Table 4: Robustness to preference reversal. We evaluate model performance ($ACC_{jud}$, %) on the original benchmarks (Normal) versus benchmarks where all preference pairs have been reversed (Reverse). Subscripts in the "Reverse" columns indicate the performance change relative to the "Normal" setting (results from Table 1).

| Method | Extended Reasoning | ALIGNX$_{test}$ | | P-SOUPS | |
| --- | --- | --- | --- | --- | --- |
| | | Normal | Reverse | Normal | Reverse |
| $\mathcal{E}$ | N/A | 50.33 | $48.67_{-1.7}$ | 38.12 | $36.57_{-1.6}$ |
| **Golden Preference** | N/A | 64.63 | *$61.83_{-2.8}$* | 81.20 | *$67.42_{-13.8}$* |
| **Qwen2.5-7B-Instruct** | ✗ | 56.33 | $47.27_{-9.1}$ | 59.33 | $68.33_{+9.0}$ |
| **DS-R1-Distill-Qwen-7B** | ✓ | 57.63 | $53.40_{-4.2}$ | 51.22 | $67.83_{+16.6}$ |
| **DeepSeek-R1-671B** | ✓ | *70.47* | *$61.53_{-8.9}$* | 70.44 | $73.33_{+2.9}$ |
| **ALIGNXPLORE-7B** | ✓ | **65.33** | **$62.13_{-3.2}$** | **62.61** | **$71.27_{+8.6}$** |

via incremental refinement without the cost of reprocessing larger signal sets. Unless stated otherwise, subsequent experiments use the base setting with $T = 4$.

**Ablation studies on training stages**    We performed an ablation study to isolate the impact of our cold-start and subsequent RL training stages. As shown in Table 1, both stages contribute positively, but RL plays a more critical role. On the in-domain ALIGNX$_{test}$ benchmark, ablating the RL stage (w/o RL) causes a more significant performance drop (65.33% → 61.80%) compared to ablating the cold-start stage (w/o Cold-start, 62.80%). The importance of RL is even more pronounced on the out-of-domain P-SOUPS dataset, particularly for nuanced dimensions like "Style," where performance plummets without RL. These results suggest that while cold-start is effective for initializing the model with basic reasoning patterns, direct reward optimization via RL is essential for refining these capabilities and enabling strong generalization.

**Online evaluation**    Using GPT-4 as a judge for pairwise comparison of personalized response generation conditioned on the generated preference descriptions, Table 2 shows that ALIGNXPLORE-7B achieves competitive win rates against baselines on both in-domain and out-of-domain scenarios, further validating its effectiveness in preference inference.

### 4.3 GENERALIZATION ABILITY ASSESSMENT

Table 3 reports the generalization performance from both input and output perspectives: **(1) Input-form generalization.** To simulate real-world scenarios, we replace standard preference pairs in the input with unstructured user-generated content (UGC, e.g., reviews). ALIGNXPLORE shows strong generalization to this new input format, achieving 61.97% accuracy and significantly outperforming all baselines. **(2) Cross-Model Generalization.** We test the transferability of the inferred preferences by using them to personalize different downstream judging models. ALIGNXPLORE again demonstrates robust generalization, consistently outperforming comparable models. We attribute this to our extended reasoning mechanism, which learns fundamental, model-agnostic preference patterns rather than surface-level correlations, leading to more portable preference descriptions.

## 4.4 ROBUSTNESS ASSESSMENT

We assess the model's robustness to shifts in user preferences through three experiments. **(1) Preference reversal.** We first test robustness by reversing all preference pairs (e.g., $y_w \succ y_l \rightarrow y_w \prec y_l$) (Li et al., 2025) to check if the model learns true inference patterns rather than dataset biases. Table 4 shows that ALIGNX-PLORE maintains strong performance, outperforming same-sized baselines and golden preferences, and competing with DeepSeek-R1-671B. This suggests our model flexibly adapts to preference patterns. **(2) Preference evolution.** We further investigate a more realistic and challenging scenario where user preferences evolve over time. Specifically, we use 8 preference pairs per user and progressively reverse the earliest ones.

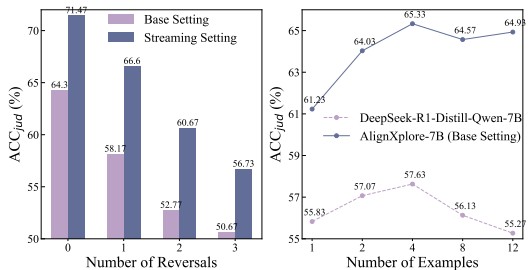

Figure 3: **Left**: ACC$_{jud}$ of ALIGNXPLORE-7B on ALIGNX$_{test}$ as user preference reverses over time. **Right**: ACC$_{jud}$ of DeepSeek-R1-Distill-Qwen-7B and ALIGNXPLORE-7B under the base setting as the number of behavioral signals increases.

Figure 3 (left) shows that our streaming mechanism consistently outperforms the non-streaming baseline across all levels of preference shift. This highlights its superior ability to adapt to temporal changes by not being confounded by outdated, inconsistent signals. **(3) Sensitivity to signal count.** We also evaluate performance as the number of behavioral signals varies from 1 to 12. As shown in Figure 3 (right), ALIGNXPLORE infers preferences accurately from just a single signal (61.23% vs. backbone's 55.83%). The performance of both ALIGNXPLORE and the backbone begins to degrade after four examples. It may be because the context becomes lengthy and saturated with noise as more signals are added. This challenge is precisely what motivates our streaming mechanism, which circumvents this long-context degradation by processing signals iteratively (see §4.5), thus maintaining stable and high performance as user history grows.

Table 5: Comparison of reward functions. $R_{jud}$ (used in all previous experiments) and $R_{gen}$ denote rewards from preference judging and response generation, respectively.

| Method | Extended Reasoning | ALIGNX$_{test}$ | | P-SOUPS | |
|---|---|---|---|---|---|
| | | ACC$_{jud}$ | ACC$_{gen}$ | ACC$_{jud}$ | ACC$_{gen}$ |
| $\mathcal{E}$ | N/A | 50.33 | 48.13 | 38.12 | 69.49 |
| **DS-R1-Distill-Qwen-7B** | ✓ | 57.63 | 48.60 | 51.22 | 69.87 |
| **ALIGNXPLORE-7B** ($R_{jud}$) | ✓ | **65.33** | 49.30 | **62.61** | **78.98** |
| **ALIGNXPLORE-7B** ($R_{gen}$) | ✓ | 61.67 | **49.40** | 56.94 | 71.82 |

## 4.5 EFFICIENCY ASSESSMENT

We evaluate the computational efficiency of our streaming mechanism as behavioral signals accumulate. In our experiment, we incrementally add 4 new signals per round and measure the average inference time and accuracy on ALIGNX$_{test}$. As shown in Figure 4, the base setting shows a significant increase in inference time as it must reprocess the entire history. Its accuracy also drops sharply at 16 signals (round 4) due to long-context challenges. In contrast, our streaming setting maintains both stable inference time and consistent performance by only processing new signals and the compact historical preference summary. This demonstrates our method's superior efficiency and effectiveness for handling growing user histories.

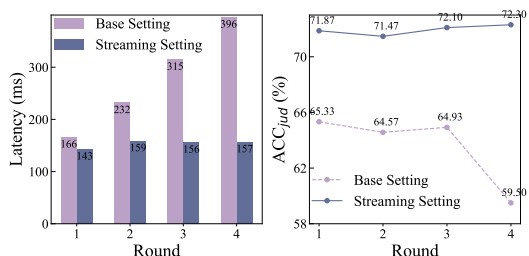

Figure 4: The average inference latency per example (**Left**) and ACC$_{jud}$ score (**Right**) of ALIGNXPLORE-7B on ALIGNX$_{test}$ as behavioral signals accumulate. Starting from 4 signals (Round=1), we add 4 new signals in each round.

## 4.6 FURTHER ANALYSIS

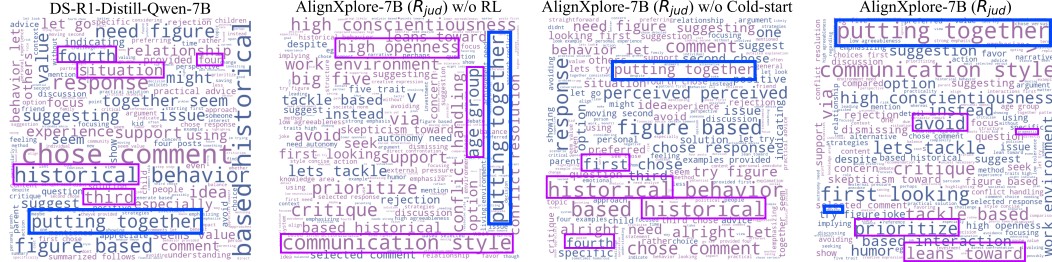

Figure 6: Word clouds of generated preference descriptions on ALIGNXtest. Terms in bounding boxes represent frequently occurring words characterizing each model's generation patterns.

We further analyze two aspects: the impact of different reward functions (**Finding 1**) and the progressive quality enhancement from our two-stage training (**Finding 2**). Appendix B.4 and B.9 provide additional analysis on RL dynamics and case studies.

**Finding 1: Judging-based rewards are superior to generation-based ones.** We compare models trained with a judging-based reward ($R_{\text{jud}}$) versus a generation-based reward ($R_{\text{gen}}$). Table 5 shows that $R_{\text{jud}}$ is better across most metrics, even including ACC$_{\text{gen}}$, suggesting that accurate preference inference naturally facilitates better personalized generation. This superiority stems from more informative training signals: Figure 5 shows $R_{\text{jud}}$

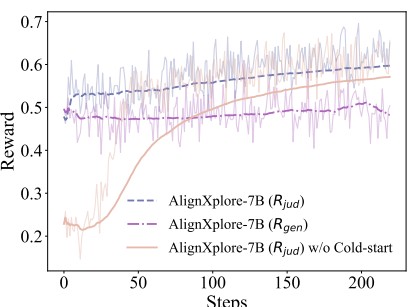

Figure 5: RL training curves with different reward functions.

enables steady learning, while $R_{\text{gen}}$ fluctuates randomly. $R_{\text{gen}}$ is ineffective due to (1) confounding factors in response probability (e.g., fluency, length) and (2) noise from using offline responses. In contrast, $R_{\text{jud}}$ provides direct, stable feedback on preference understanding.

**Finding 2: Two-stage training progressively refines preference descriptions.** Figure 6 illustrates how our training stages progressively enhance description quality. The backbone model produces only general terms (e.g., "historical," "situation"). Cold-start training introduces specific preference dimensions (e.g., "communication style," "age group") but lacks synthesis. RL alone offers limited improvement, focusing on generic concepts like "helpfulness." In contrast, the combined two-stage approach yields actionable guidance with diverse dimensions and concrete actions (e.g., "avoid," "prioritize," "leans toward"). This evolution from general observations to specific, actionable hypotheses mirrors human inductive reasoning (Heit, 2000; Fränken et al., 2022) and is naturally encouraged by our framework without explicit supervision.

## 5 CONCLUSION

This work presents the first systematic investigation of extended inductive reasoning in LLMs for personalized preference inference. Our model, ALIGNXPLORE, demonstrates that extended reasoning can effectively bridge the gap between implicit behavioral signals and explicit preferences. Comprehensive experiments show that ALIGNXPLORE achieves superior personalized alignment while maintaining strong efficiency, generalization, and robustness. The success of our two-stage training strategy provides valuable insights into developing LLMs' inductive reasoning capabilities, suggesting that combining cold-start with RL can effectively guide models to learn generalizable reasoning patterns. Our findings also reveal several promising directions for future research, such as extending the success of our approach in preference inference to other inductive reasoning tasks, such as scientific hypothesis generation and pattern discovery in unstructured data.

## ETHICS STATEMENT

This work enhances the preference inference capability of models, enabling them to better serve human users by understanding and responding to their individual preferences. However, it may involve potential risks related to user privacy and bias. By inferring personalized preferences, there is a possibility of inadvertently amplifying existing biases in the data or misinterpreting user intent. To mitigate these risks, we ensure that our approach incorporates robust fairness and transparency measures. We also prioritize user consent and implement mechanisms to ensure that user data is anonymized and securely handled. Furthermore, we encourage ongoing monitoring of the model's performance in real-world scenarios to identify and address any unintended consequences, thus ensuring that the model's deployment remains ethical and aligned with user interests.

## REPRODUCIBILITY STATEMENT

Our code is publicly available in an anonymized repository `https://anonymous.4open.science/r/ICLR2026-AlignXplore`, with detailed running instructions in the Readme. The training and testing setups are specified in §4.1 and Appendix B.1 to guarantee full reproducibility.

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

## A   USE OF LLMS

We use LLMs for two purposes. First, we use LLMs to generate cold-start training data, which serves as a high-quality alternative when manual annotation is prohibitively expensive (our method is detailed in §3.3 and Appendix C). Second, we use LLMs for online evaluation to assess the quality of model-generated responses. This approach, using GPT-4 judgments as a proxy for human evaluation, is a widely adopted practice. It is supported by benchmarks like AlpacaEval (Dubois et al., 2024), which report high agreement with human annotations, and has been successfully employed in numerous influential studies on preference learning and model alignment (Liu et al., 2023; Gu et al., 2024). The specific prompt for this evaluation is provided in Appendix C.

Table 6: Summary of evaluation benchmarks. For preference directions, ↑ and ↓ represent preferred and non-preferred examples, respectively, with their quantities shown in parentheses. The "In-domain" column (✓/✗) indicates whether the benchmark's preference dimensions are seen during training.

| Benchmark | Dimensions and #Examples | In-domain |
|---|---|---|
| ALIGNX$_{\text{test}}$ | 90 preference dimensions (3,000 examples in total, ∼1:1 ratio for ↑/↓ preferences) | ✓ |
| P-SOUPS | "Expertise" (↑: 300, ↓: 300); "Informativeness" (↑: 300, ↓: 300); "Style" (↑: 300, ↓: 300) | ✗ |

## B EXPERIMENTS

### B.1 IMPLEMENTATION DETAILS

Our training and test sets are derived from ALIGNX, which proposes a 90-dimensional preference space (incorporating universal values, basic human needs, and prevalent interest tags). The dataset utilizes forum interactions and human-LLM interactions to construct 1.3 million examples, making it currently the largest and most comprehensive dataset for personalized alignment. However, preference signals in the original user interactions are relatively sparse, which previously hindered effective preference inference. To address this issue, we introduce a refined data construction approach. Specifically, we ensure that each target pair is associated with at least five preference dimensions, where all interaction history demonstrates consistent, non-neutral preference directions, while avoiding conflicting preferences across other dimensions. We constructed 10,000 data entries containing only "pair-wise comparative feedback" as interaction history, with 7,000 used for training and 3,000 for testing. When 7,000 instances are used for cold-start training, we select 3,980 instances for the first round and 5,278 instances for the second round based on $R(r, d) = 1$. Additionally, we constructed 3,000 entries containing only "user-generated content" as interaction history for generalization validation.

The training is conducted on 8 NVIDIA A100 GPUs using Adam optimizer (Kingma, 2014), with DeepSpeed ZeRO-3 (Rajbhandari et al., 2020) and Flash-attention-2 (Dao, 2023) for optimization. We employ the following hyperparameter configuration: learning rate of 1e-6, 50 warmup steps, 4 training epochs, and maximum prompt/generation lengths of 8,192/2,048 tokens. During RL, we set the mini-batch size to 128 for each step.

### B.2 BENCHMARK DETAILS

Table 6 shows the summary of the evaluation benchmarks.

### B.3 BASELINE DETAILS

We compare our approach with various baseline methods and models:

- **Directly given preference descriptions:** (1) *Null*: no preference description is provided; (2) $\mathcal{E}$: using behavioral signals directly as preference descriptions without inference; and (3) *Golden Preference*: ground-truth preference descriptions provided by the benchmark. Note that golden preference descriptions, while semantically accurate, may not necessarily lead to optimal downstream personalization performance due to potential gaps in model compatibility.

- **Previous specialized methods for inductive reasoning and personalization:** (1) *LMInductReason* (Qiu et al., 2024) employs iterative hypothesis refinement to enhance LLMs' inductive reasoning capabilities; (2) *VPL* (Poddar et al., 2024) introduces latent variables to model individual preferences; (3) *EXPO* (Hu et al., 2025b) improves upon DPO by introducing an explicit objective function, which provably avoids sub-optimal behaviors of implicit reward modeling; and (4) *PBA* (Li et al., 2025) maps behavioral examples to structured preference scores along predefined dimensions, then converts them to natural language descriptions.

- **Preference descriptions generated by state-of-the-art LLMs:** The LLMs range from small-sized models including *Qwen2.5-7B-Instruct* (Team, 2024) and *DS-R1-Distill-Qwen-7B* (DeepSeek-AI, 2025), to large-sized models including *QwQ-32B* (Team, 2025), *Qwen3-32B* (Yang et al., 2025),

Table 7: $\text{ACC}_{\text{jud}}$ of ALIGNXPLORE-7B when reversing the first preference pair of the user and keeping the later pairs (and test pairs) consistent with the final preference. **Extended Reasoning**: whether the model generates preference descriptions with extended reasoning. $T$ refers to the number of examples in $\mathcal{E}$ in both training and inference. $\hat{d}$ indicates whether historical preferences are empty or not.

| Method | Extended Reasoning | Setting | $T$ | $\hat{d}$ | ALIGNX$_{\text{test}}$ | P-SOUPS | | |
|---|---|---|---|---|---|---|---|---|
| | | | | | | Informativeness | Style | Expertise |
| ALIGNXPLORE-7B | ✓ | Base | 8 | ✗ | 58.17 | 51.66 | 67.67 | 61.17 |
| ALIGNXPLORE-7B | ✓ | Streaming | 4 | ✓ | 66.60 | 58.97 | 69.67 | 65.67 |

*GPT-4* (Achiam et al., 2023), and *DeepSeek-R1-671B* (DeepSeek-AI, 2025). These models cover both concise reasoning and extended reasoning patterns.

Furthermore, to verify the effectiveness of our approach, we also compare with ALIGNXPLORE-7B w/o RL and w/o Cold-start under the base setting, which only uses cold-start training and RL for preference inference, respectively.

For VPL (Poddar et al., 2024) and EXPO (Hu et al., 2025b), we train 4 epochs on DeepSeek-R1-Distill-Qwen-7B using $\mathcal{D}_{\text{rl}}$. Note that VPL employs its own specialized downstream model for preference-guided judgment. For other baselines, we generate roles or preferences using the corresponding models and input them into Qwen2.5-7B-Instruct for evaluation. LMInductReason (Qiu et al., 2024) follows the original paper's implementation, where content generation is replaced by Qwen2.5-7B-Instruct. After iteratively generating rules, the final rule is provided to Qwen2.5-7B-Instruct to generate preference selections. PBA (Li et al., 2025) uses the method from the original paper to extract consistent preferences from the interaction history of each benchmark.

### B.4 LENGTH EVOLUTION

We present the changes in generation length during the reinforcement learning process for ALIGNX-PLORE-7B ($R_{\text{jud}}$) and ALIGNXPLORE-7B ($R_{\text{gen}}$) in Figure 7. As training progresses, the average generation length of the model continuously decreases. Our analysis suggests that, due to cold-start training, although the model is guided to analyze the appropriate preference dimensions, it tends to repetitively reproduce content from the behavioral signals, with low confidence in the analysis and many redundant and fluctuating dimensional interpretations. After reinforcement learning, the model's analysis direction

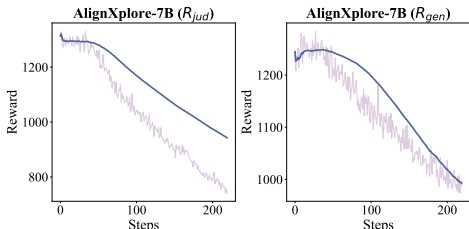

Figure 7: Curves of generation length for ALIGNXPLORE-7B with different reward functions during RL training.

becomes clearer. For preference interpretation of behavioral signals, the model now only mentions key terms that reflect preferences, enabling it to quickly analyze and summarize user preferences. This aligns with the analysis presented in §4.6.

### B.5 ROBUSTNESS ASSESSMENT

Figure 7 shows the performance of ALIGNXPLORE-7B on different datasets under the base setting and streaming setting when 8 preference pairs are given for each user and the first preference pair is reversed. Since the streaming inference mechanism allows the model to refine preference descriptions during streaming inference of user preferences, it demonstrates robustness and generalization when facing inconsistent or time-varying user behavior preferences. As a result, it consistently outperforms the base setting on both in-domain and out-of-domain datasets.

Table 8: $\text{ACC}_{\text{jud}}$ of different models with Qwen3-8B as the backbone.

| Model | $\text{ALIGNX}_{\text{test}}$ | P-SOUPS |
|---|---|---|
| **Qwen3-8Bnon-thinking** | 59.10 | 62.77 |
| **Qwen3-8Bthinking** | 67.87 | 66.43 |
| **ALIGNXPLORE-8B** | 69.37 | 69.10 |

Table 9: Training costs of different methods.

| Model | VPL | PBA | ALIGNXPLORE |
|---|---|---|---|
| **Time** | 2h | 0.2h | 4h |

## B.6 ANALYSIS OF DIFFERENT BACKBONE MODELS

To demonstrate the generalizability of our approach across different backbone models, we conduct additional experiments using Qwen3-8B (Yang et al., 2025) with thinking mode as the backbone. Table 8 shows that ALIGNXPLORE consistently improves performance over the backbone model under the base setting. This demonstrates that our approach's effectiveness is not limited to a specific backbone architecture but generalizes well across different model foundations.

## B.7 COMPUTATIONAL COST ANALYSIS

We estimate the one-epoch training time for various methods on 7,000 samples using 16 H20 GPUs. This estimate is based on training DeepSeek-R1-Distill-Qwen-7B for VPL and ALIGNXPLORE and Llama-3.1-8B-Instruct for PBA (as reported in its original paper (Li et al., 2025)). Table 9 shows the training costs for different methods. Our method comprises two stages: cold-start training, which requires 0.3 hours per epoch on 16 H20 GPUs, and reinforcement learning, which requires 3.7 hours per epoch on the same hardware. We consider this training time worthwhile because the two-stage approach yields significantly better results, and the more time-consuming RL stage is crucial for enhancing the model's preference inference capability. Furthermore, Table 10 shows that the inference cost for the two-stage model is comparable to that of the RL-only model and significantly lower than that of the cold-start-only model. Therefore, the two-stage approach does not increase complexity or resource requirements during inference.

## B.8 SENSITIVITY OF ALIGNXPLORE-7B TO THE QUALITY AND DIVERSITY OF $\mathcal{D}_{\text{COLD}}$

The primary role of cold-start training is to help the model learn to analyze specific and fine-grained preference dimensions, enabling a more systematic analysis. The word cloud in Figure 6 for ALIGNXPLORE-7B w/o RL shows the emergence of specific dimensions like "communication style" and "age group," which are absent before this stage. However, the main improvement in preference inference capability comes from the reinforcement learning (RL) stage. The ablation study in Table 1 shows that removing RL causes a more significant performance degradation than removing cold-start training. Specifically, ALIGNXPLORE-7B w/o Cold-start still outperforms the base model (DeepSeek-R1-Distill-Qwen-7B) by 8.12%, whereas ALIGNXPLORE-7B w/o RL shows only a 4.29% improvement. This indicates that the quality of the synthetic data is not the decisive factor for ALIGNXPLORE performance.

Table 10: Inference costs of different stages.

| Model | ALIGNXPLORE-7B | ALIGNXPLORE-7B w/o Cold-start | ALIGNXPLORE-7B w/o RL |
|---|---|---|---|
| **Time** | 157ms | 100ms | 230ms |

Table 11: ACC$_{jud}$ on low-quality, low-diversity data.

| Model | ALIGNX$_{test}$ | P-SOUPS |
|---|---|---|
| **DeepSeek-R1-Distill-Qwen-7B** | 57.63 | 51.22 |
| **ALIGNXPLORE-7B w/o RL (Low quality and diversity)** | 56.87 | 51.11 |
| **ALIGNXPLORE-7B (Low quality and diversity)** | 62.90 | 59.11 |
| **ALIGNXPLORE-7B w/o RL** | 61.80 | 55.55 |
| **ALIGNXPLORE-7B** | 65.33 | 62.61 |

Furthermore, we conduct an experiment to directly investigate ALIGNXPLORE's sensitivity to the quality and diversity of the synthetic data. To degrade the training data, we randomly select 1,000 samples and regenerate their reasoning chains and preferences using a much weaker teacher model (DeepSeek-R1-Distill-Qwen-7B). The results in Table 11 show that after cold-start training with this lower-quality and less diverse data, ALIGNXPLORE performs even worse than the base model (DeepSeek-R1-Distill-Qwen-7B). However, with subsequent reinforcement learning, it performs far better than the base model, and its performance drops only slightly compared to the standard ALIGNXPLORE. This demonstrates our method's robustness to the quality and diversity of the synthetic data, while also indicating that cold-start training provides limited improvement to the model's preference inference ability.

## B.9 CASE STUDY

DS-R1-Distill-Qwen-7B tends to be more general and one-sided when analyzing preferences from behavioral signals, which may lead to the omission of important points during the analysis. After cold-start training, ALIGNXPLORE-7B w/o RL provides more comprehensive and systematic analysis of the preference dimensions, but expressions indicating uncertainty, such as "?" and "Not clear yet," frequently appear, along with extensive repetitions of content from the behavioral signals, such as "User describes facing harassment by a host due to his identity." After reinforcement learning, these are replaced by more confident statements and clearer analyses, indicating that RL significantly aids in making inductive reasoning more precise and focused. In the streaming setting, behaviors that contrast with and adjust according to historical preferences may emerge, such as "looking at the past preferences," "fits," and "consistently."

For non-extended-reasoning models (e.g., Qwen2.5-7B-Instruct), the preference descriptions are provided directly. However, due to the lack of reasoning processes, some unreasonable preference descriptions emerge. In fact, during the analysis, the model focuses more on the user's responses or the user's tendencies toward different responses, rather than focusing on the content of the questions. However, many of the analyses provided by Qwen2.5-7B-Instruct are based on the content of the questions, such as "Interest in Personal Development and Self-Improvement."

---

**Prompt for Case Study**

A conversation between User and Assistant. The User asks a question, and the Assistant solves it. The Assistant first thinks about the reasoning process in the mind and then provides the User with the answer. The reasoning process is enclosed within `<think> </think>` and answer is enclosed within `<answer> </answer>` tags, respectively, i.e., `<think> reasoning process here </think> <answer> answer here </answer>`.
**User:** You must put your answer inside `<answer> </answer>` tags, i.e., `<answer> answer here </answer>`.
**This is the problem:**
Generate the user's preference based on their historical behavior.
**This person has chosen or rejected comments on some posts:**

    1. **Post:**
       Sorry for format on mobile etc.

---

My girlfriend[22] and I[22] decided to go away somewhat last minute. It's our first trip together. We're away in France, not far from Lille. We decided to get an apartment on Airbnb, it was inexpensive and so beautiful. It was perfect.

Except, and this is the very concise version, we show up and the host looks surprised to see us. I speak fluent french and my girlfriend doesn't, and he doesn't speak any English so I held the conversation even though she handled all the booking and liaising. He kept on asking if my girlfriend was the one in the picture as he was expecting the male and female in the picture. Her picture was an old one of her and her friend. I tried explaining this to him and he acts as if I'm not understanding french, my own first language, properly.

Turns out that my girlfriend had arranged for a bouquet for me which was waiting for me in the main room. He couldn't wrap his head around it. He was acting somewhat civilised until I saw the flowers and hugged my girlfriend. Then he pieced it together and started acting hostile. His tone changed and he started laying down impromptu ground rules like no liquids near the sofa and no shoes on the rug and don't trash the place, return it how you got it. All while his tone got more and more hostile.

It's his apartment and he probably has a key and I just feel so unsafe. I'll get over it once I process it I'm sure but I'm literally just starting medication for depression and anxiety and being at home is so comforting for me, which is why I was looking forward to getting in after travelling for what felt like eternity. It just fucking sucks that I can't go away with my significant other without feeling unsafe and vulnerable. And I feel so bad cus my girlfriend is trying so hard to make this the best trip and she feels bad that our host is a homophobe even tho that's literally nothing to do with her.

Idk if this made sense I'm tired and sleep deprived but can't bring myself to sleep. Please send some kind gay words so I don't feel like the only gay in the world. Little pick me up please.

**Chosen:**
I live 20mins from Lille! (In Belgium near the Belgian/French border)
I've heard before that people near the border/north of France seem to be more conservative.
I'm sorry this happened to you.

**Rejected:**
Un bon gros fdp en somme.

2. **Post:**
So, I'm a 21yo bisexual (maybe pansexual? I'm completely open to it but haven't really met or talked to someone to know), living in the Chicago area. I realized and have been out as bisexual since I was 17. I've slept with women, and I hate to admit it because of the 5 I've been with, only 2 were legitimately into girls, the rest was just drunken young adult barsexual experimenting. This is already sounding great right? *sarcasm* Well, I've dated my fair share of men and quite honestly I'm bored. I have yet to find a girl that there is a mutual interest. 2 or 3 girls have liked me, but I honestly just wasn't into them in a romantic way, and that was all they wanted, no friendship. I really want to date a girl already! I have no clue how to flirt with girls, I can never tell when I'm being flirted with, or how to tell them I've been out as bisexual for almost 5 years, have never dated a girl but screwed a bunch. Doesn't sound too good on my end. I kind of have a crush on a bisexual friend of mine and would like to try for her, but I've tried and failed at flirting or catching her attention in more than a friendly way. I have no clue what to do unless I'm in bed with them. I have no idea how to flirt with girls. Send help.

**Chosen:**
I wish I could offer some good advice, but I'm not so good at flirting myself.
I wish you the bestest of luck.

**Rejected:**
I flirt like a pro and I can be smoother than butter... But I have no idea how to articulate the HOW aspect of it. I really want to help, but I'm grasping for words

here.

*code-sloth ponders a moment, as a sloth does when stumped*

Witty banter and humor are my main skills, which take a bit to learn for each gal's tastes. At the very least, be genuine and sincere. Whether you're a smooth talker or as mentally coordinated as a deer on ice, sincerity goes a long way to endearment. Thoughtful, sincere, genuine. Start there, then you can develop your own style of flirting.

Oh! And be aware of when to shut up or back off. That great one-liner she lined you up for? Don't always blurt it out. She's not reacting well to certain things (usually dirty lines)? Cool your jets and mellow out. It's a balancing act, and you'll pick up on it over time.

Sometimes you won't even need to flirt actively. Talking about a mutual subject (I love video games, for example) can cause the "oh, I dig this chick" feeling that flirting does. I'm not above the whole "Can this controller fit between your boobs?" line of discussion, but you don't have to be on your witty toes ALL the time. Would you date a good yet incompatible flirt or someone you shared a common interest with? Probably the latter.

Regarding pickup lines: No. Don't use them in serious context. Jokingly yes, but don't play that card on the table first.

Woah, that got a bit verbose. Sorry!

3. **Post:**

As a 20 year old, it made me sad to see so many of you calling yourself old! Not that that's a bad thing. I don't think teenage/20s years are the peak of your life. I was having this conversation with my ex girlfriend (yeah...I know) the other day and she said this is a really shitty confusing time and IA. and besides I have so many health issues, I'm looking forward to having surgery and stabilising and having more of a grip on my life/mental stability in my 30's and 40's and all the years after that.

Anyway this thread is kind of OT but to all the 30+ ladies on here, you're not old and even when you are you can still be a badass. I know 70 year olds that are kicking ass and when you think about it they were born in the 1940's, and healthcare sucked then.

Saw a thread here asking for members over 30...

**Chosen:**

Thanks, thats nice of you to say :)

**Rejected:**

It's cool. I'm immortal anyway. Which reminds me, I have this sword fight I need to get to...

*Queen Plays*

4. **Post:**

My boss, after ghosting me for two weeks, fired me over text this morning. She was fine when I came out to her, but after discovering that I'd actually take medication to change my body, she's hated me. I worked with kids so she was always afraid I'd corrupt their minds. As far as I know, parents didn't care or know. They assumed me to be female and their kids assumed me to be male but otherwise it was all fine.

I guess I'll stay with my other job and wait for them to start paying me (still a volunteer until the fall when a job opens up). At least I'm out there and everyone cool with it (behavioral health center).

Should I just stay closeted in the future? I guess I will until I pass and then I'd be stealth? Fired for "inability to keep up with changes"

**Chosen:**

When are we going to get laws protecting LGBT people from getting fired? How come we didn't get it yet? I'm so sorry this happened to you.

**Rejected:**

If your employer has not been documenting these "failures," then they are not in a good position legally. The EEOC (equal employment opportunity commission) is a federal agency that investigates discrimination due to gender identity. You can and

should contest your firing; check them out online today. You have two weeks from
firing to contest. Please do.

It's possible that nothing will come of it, but very often, they can find other workers
who will substantiate your manager's motives. You will have done the right thing by
putting it into their hands.

Obviously, you have to assess your own emotional needs before taking this step.
Quick action is also important for unemployment, which is the right of any worker
released for no fault of their own. You will have to contest the firing with the
unemployment office, too. Your manager is cheating you financially.

Keep your head up. If you choose, this is new work, more important than any you
have done before. Your task is to try to get justice for yourself and other trans people.
Just to try. That is more than enough.

It's great of you to volunteer at the behavioral health center! I hope your career there
will be rewarding.

**Assistant:** `<think>`

---

**Preference Generated by DS-R1-Distill-Qwen-7B**

Okay, so I need to figure out the user's preference based on their historical behavior. They've
provided four examples where they either chose or rejected comments on posts. Let me
analyze each one step by step.

**First Post:** The user shared a personal anecdote about traveling with a partner and facing a
negative host due to language and cultural barriers. The user felt unsafe and lost depression
and anxiety. They received a chosen comment that was empathetic and a rejected one that was
dismissive. From this, it seems like the user appreciates genuine support and understanding,
especially in sensitive situations.

**Second Post:** The user identified as a bis (or pan) and discussed dating experiences. They
felt unsuccessful and frustrated. The chosen comment was a friendly acknowledgment, while
the rejected one was more detailed but still empathetic. The user values honesty and being
heard, especially in personal and vulnerable contexts.

**Third Post:** The user shared about facing termination due to their gender identity, feeling
misunderstood and unhelped. The chosen comment was a compliment, and the rejected
comment was more informative but still empathetic. The user seems to thrive on positive
reinforcement and feels supported when others acknowledge their strengths.

**Fourth Post:** Another situation where the user faced professional discrimination. They were
fired for being closeted and chose a comment that expressed solidarity and disappointment,
while the rejected comment provided practical steps for legal action. The user values solidarity
and stands with others in similar situations but also seeks practical advice.

Looking at these, the user seems to prefer comments that are empathetic, supportive, and
acknowledge their struggles. They appreciate genuine concern and positive reinforcement.
They also seem to seek practical advice when applicable, especially in situations where they
feel discriminated or unsafe. However, they don't seem to appreciate overly dismissive or
critical comments, even if they're well-meaning.

So, the user's preference is for comments that show understanding, support, and positivity.
They value empathy and practical help when needed, but they don't like being dismissed or
told to "get over it." They likely seek validation, reassurance, and encouragement in their
challenging situations.
`</think>`

The user's preferences are evident in their analysis of past interactions, where they favor
comments that demonstrate empathy, support, and positivity. They appreciate genuine concern
and acknowledge the user's struggles, seeking both practical advice and affirmation in their
challenging situations. They dislike overly dismissive or critical comments, even if well-
intentioned, and value solidarity and encouragement. Thus, the ideal comment for the user
would be one that shows understanding, offers support, and provides positive reinforcement
or practical advice when applicable.

**Preference Generated by ALIGNXPLORE-7B w/o RL**

Okay, let's start breaking down the information provided here. The task is to generate the user's preferences based on their historical interactions—specifically the comments they've chosen versus rejected across different social media posts. There are several factors mentioned related to psychological needs and communication styles mentioned earlier, such as openness, agreeableness, autonomy, approach to conflict resolution, etc., along with specific communication nuances.

First, looking at each example provided:

**Starting with Post #1:**

User describes facing harassment by a host due to his identity. His main issue here seems emotional distress stemming from societal norms affecting his identity, particularly gender-related discrimination. He feels unsafe, uses medication for mental health issues, seeks support, and wants validation.

He chose a response acknowledging location near a border area (which might indicate regional context?), noting potential conservatism elsewhere, expressing sympathy, and thanking him. Rejected a vague remark (*un bon gros fdp en somme*), suggesting the user prefers empathetic responses rather than dismissive ones.

So possible indicators include High Openness (to experience/emotional intelligence?), Low Agreeableness since he's upset, High Need for Autonomy—he wants control over his situation (e.g., staying home, dealing with anxiety independently?), Conflict Handling – confrontational or avoidance? Not clear yet.

**Moving onto Post #2:**

User identifies as bisexual (maybe pan), struggling with attraction despite previous experiences. Struggles finding authentic romantic partners beyond casual encounters. Wants advice but feels frustrated. Chooses a reply offering sympathy and encouragement ("bestest luck"), rejects a lengthy, verbose guide filled with witty tips. That suggests preferring brevity over detail-heavy solutions—maybe valuing conciseness vs thorough explanations.

Possible clues: Low openness to complex strategies, higher need for autonomy in handling situations themselves versus relying on external methods. Also, rejects overly structured approaches preferring direct empathy.

**Third Interaction (Post#3):**

User celebrates aging positively, mentions achieving maturity, health challenges, and future goals. Chose a simple thank-you response instead of playful/humor-oriented replies referencing swords/fighting. Indicates seriousness regarding aging positively, dislikes sarcasm or flippant replies.

Shows emphasis on sincerity/respectful communication avoiding trivialization. Emphasizes authenticity and substance over levity.

**Fourth Scenario (Boss situation):**

Emotionally impacted by losing job over gender identity concerns. Selected compassionate question about legislation protection plus sympathy ("how come?"), whereas the rejected comment gives legal/procedural guidance including steps to contest via agencies and unemployment offices. The user chooses emotional expression over procedural advice. Suggests prioritizes emotional support and communal solidarity over practical steps. Prioritizes feeling validated and supported rather than facing bureaucracy head-on immediately.

So here autonomy comes in – user prefers staying informed but seeks immediate comfort/support rather than taking proactive steps immediately. Maybe prefers empathetic validation over practical action plans.

**Now compiling these observations:**

**Looking at Personality Traits:**

- **Big-Five Openness:** Lower probably. Because they reject verbose/detailed answers in favor of shorter/more straightforward suggestions. Unless the rejection is specific to inappropriate content... Not sure though. Alternatively, openness relates to creativity/exploration—maybe limited here.

    Wait, but in Post #1, the rejection of "un bon gros fdp en somme" (which roughly translates to 'f*** off') shows aversion to crude language, preferring respectful tone. Maybe mid-openness? Hmm tricky.

- **Agreeableness:** Likely higher than average given tendency to choose comforting/empathic remarks over confrontational or sarcastic ones. However, in fourth instance, the chosen comment asks 'how come' implying empathy but also acknowledges systemic issues (laws?), indicating balance between supportive and critical perspectives? Or perhaps moderate agreeableness.

- **Need For Autonomy:** Seems strong—they prefer independent solutions, resisting dependence on others' input (as seen in second post rejecting elaborate tips). Though in fourth scenario, they sought help from authorities (law/companies?) but still chose emotional affirmation over procedural steps. Maybe balancing autonomy with seeking expert resources selectively.

- **Conflict Handling Approach:** From first and fourth cases, the user reacts strongly emotionally (anxiety, anger), indicating avoidance of direct confrontation unless necessary. Preferred responses offer reassurance rather than addressing root causes directly. Thus prefer indirect/empathic resolutions.

- **Communication Style:** From examples above, the user tends toward concise messages avoiding excessive details. Rejected options included lengthy guides filled with anecdotes/examples. Hence leans towards Concise/Prefer direct communication over elaboration.

- **Work Environment Preferences:** Uncertain yet. Third post mentions volunteering positively. Fourth talks about employment issues. Possibly values environments allowing flexibility and autonomy, avoiding rigid structures that led to firing. Prefers places where boundaries respected ethically (maybe corporate policies aligned with LGBTQ+ rights?).

- **Workplace Needs:** Likely values transparent workplaces adhering to anti-discrimination laws, thus advocating for legal protections. Also wants safe spaces free from hostile environments (as per first story).

- **Achievement Needs:** In third scenario celebrates maturity and long-term objectives (ageing positively), showing appreciation for progress and stability. Achievement motivation exists but framed positively (not competitive, but attaining inner peace/stability).

- **Autonomy Needs:** Strong. Avoid relying on others' input (rejected tips in #2), prefers making own decisions guided by empathy (e.g., trusting their instincts in dating despite uncertainties). Want control over decisions, reduce dependency on external validation/suggestions.

- **Law Awareness:** Fourth scenario acknowledges potential lack thereof affecting workplace fairness. Suggests awareness/careful consideration regarding legislation. May value informed decisions backed by legal insights but balance with emotional responses.

**Putting together all threads...**
**Finalizing categories:**
**Big Five:**

- **Low/Openness?**: Since rejects complex advice, leans toward simpler expressions. However, openness relates to imagination/intellect—not sure. Maybe neutral/mid-range?

- **Moderate-High Agreeableness**: Chooses supportive tones, avoids hostility. Empathetic nature evident throughout choices.

**Needs:**

- **Strong Autonomy**: Prefers independent judgment, resists external influences (rejection of structured guides in dating, seeking validation instead of proactive measures).

- **Need for Achievement**: Mixed signals—but focuses on overcoming anxieties via proactive actions (getting back mentally).

- **Workplace Safety**: Prioritizes environments respectful of identities, advocating policies preventing discrimination.

- **Conflict Management**: Prefer mediating through dialogue/acknowledging issues indirectly rather than combative methods.

**Communication Style:** Concise, avoids unnecessary elaboration. Focus on clarity and sincerity.

**Law:** Shows awareness of systemic factors impacting employment/freedom, willing to challenge injustices but seeking respectful dialogues.

Thus synthesizing all points into coherent preferences...

`<think>`

`<answer>`

Based on the analyzed behaviors, the user exhibits the following preferences derived from their interactions:

1. **Personality Traits (Big Five):**
   - **Openness:** Moderately low/high. Prefers pragmatic solutions over elaborate suggestions (rejects verbose advice but values empathy).
   - **Agreeableness:** High. Chooses supportive, compassionate language and prioritizes interpersonal harmony.

2. **Core Psychological Needs:**
   - **Need for Autonomy:** Very pronounced. Resists reliance on external advice/influences (rejects complex dating tips, favors personal intuition). Seeks self-determined pathways amidst stress/anxiety.
   - **Workplace Safety:** Strong advocacy for inclusive workplaces free from bias/discrimination.
   - **Emotional Security Over Pragmatism:** Values emotional validation and communal solidarity (choosing heartfelt condolences) over pragmatic steps (legislation guidance).

3. **Social Interaction Styles:**
   - **Directness + Empathy:** Communicates concisely yet sincerely, balancing honesty with sensitivity (chooses brief affirmation vs verbose explanations/rejections).
   - **Avoidant Conflict Resolution:** Tends to sidestep contentious topics (e.g., avoids discussing workplace discrimination head-on except when validating feelings).

4. **Legal Awareness & Personal Achievement:**
   - A conscious awareness of legal frameworks influencing life decisions (aware of EEOC protections but leans on communal solidarity).
   - Personal fulfillment tied to overcoming vulnerabilities (mental health improvement linked to traveling away for escape).

5. **Work Environment Needs:**
   - Thrives in transparent, equitable workplaces aligned with individual values (supports LGBTQ+ rights legislation advocacy).

`</answer>`

---

**Preference Generated by ALIGNXPLORE-7B under Base Setting**

Okay, let's start breaking down these interactions to figure out the user's preferences using the given criteria. First, looking at each post response:

**Starting with Post 1:** The user chose a comment acknowledging the situation politically ("people near the border") and showing empathy. They rejected a shorter phrase ("un bon gros fdp"). This suggests the user values contextual analysis and empathy rather than brevity. So maybe they prefer deeper insights over concise replies.

**Moving to Post 2:** Here, they picked a supportive reply emphasizing luck and sincerity without practical advice. Rejected a verbose explanation filled with tips. This points toward preferring brevity and warmth over extensive suggestions. Also, rejecting humor ("witty banter") implies less interest in overtly playful solutions unless sincere.

**In Post 3:** Chose gratitude instead of sarcastic remarks about immortality. Shows appreciation for genuine thanks versus edgy content. Emphasizes gratitude and positivity.

**Lastly, Post 4:** Opted for emotional support asking for societal protections vs professional guidance steps. Indicates valuing emotional validation first before actionable advice. Rejects detailed procedural guidance, focusing on feelings.

Putting together, the user likely prioritizes warmth, empathy, conciseness, and avoids overly verbose/narrative-heavy responses. Their communication leans towards direct yet compassionate messages, seeking connection through shared experiences over complex explanations. They might value authenticity and emotional support above technical details or sarcasm.
`</think>`

`<answer>`
**User Preferences Analysis Based on Historical Behavior:**
**Key Traits Identified Across Selected Responses:**
1. **Emotional Support Over Detailed Advice:** - Prefers empathetic affirmations (e.g., "Thanks, that's nice of you" vs lengthy guides on legal processes).
2. **Contextual Humor/Social Awareness:** - Chooses politically relevant observations (border regions) over blunt humor ("un bon gros fdp").
3. **Direct Empathy Without Overexplanation:** - Selects compassionate statements focusing on connection ("be sincere"), rejects verbose explanations (flirting tips, "let me help").
4. **Privacy Focus:** - Avoids overly personal/critical content, opting for platonic solidarity (volunteering at health centers, "stealth" strategy).
**Personality Traits Alignment:** - Low openness to abstract concepts (preferring straightforward empathy). - High agreeableness (prioritizing harmonious interactions). - Strong desire for **authentic communication**, disliking pretentious language or performative politeness.
`</answer>`

---

**Preference Generated by Qwen2.5-7B-Instruct**

Based on the user's historical behavior on the platform, the following preferences and tendencies can be inferred:

1. **Support for LGBTQ+ Community:**
   - The user frequently engages with posts and comments related to the LGBTQ+ community, especially regarding trans issues, relationships, and personal experiences.
   - They often choose to offer comfort, support, and validation to others going through similar struggles, showing empathy and a supportive nature.

2. **Interest in Personal Development and Self-Improvement:**
   - The user seeks advice and guidance on personal issues such as dealing with ex-partners, coming out, and navigating family and social relationships.
   - They are open to receiving and giving advice, showing a willingness to engage in meaningful conversations that can help others.

3. **Emotional Support:**
   - The user often chooses comments that provide emotional support, validation, and encouragement. They appreciate kind words and genuine responses.
   - They tend to avoid confrontational or dismissive comments, preferring to offer understanding and empathy.

4. **Sensitivity to Pronouns and Gender Issues:**
   - The user demonstrates sensitivity to pronoun usage and gender issues, often choosing comments that reflect understanding and support for trans individuals.

- They are likely to be aware of and respectful of different gender identities and pronouns.

5. **Struggle with Intimacy and Relationships:**

  - The user frequently mentions difficulties with intimacy, relationships, and sexual experiences, indicating a need for support and guidance in these areas.
  - They seem to be seeking validation and advice on how to navigate relationships, both romantic and platonic.

6. **Interest in Sexual Health and Consent:**

  - The user shows an interest in sexual health and consent, discussing sensitive topics such as sexual experiences and boundaries.
  - They value explicit and honest communication in relationships, showing a preference for clear and respectful interactions.

Based on these preferences and tendencies, it can be inferred that the user values emotional support, empathy, and genuine advice in their interactions. They are likely to appreciate content that is respectful, understanding, and supportive of their struggles, especially in relation to personal relationships, gender issues, and sexual health.

---

**Preference Generated by ALIGNXPLORE-7B under the Streaming Setting**

Okay, let me start breaking this down. The task is to generate the user's preferences based on their selected versus rejected comments across these four posts. First, I need to analyze each post individually, noting why the chosen response stands out compared to the ones they dismissed.

Starting with Post 1: Here, the author talks about dealing with a homophobic host and feeling unsafe. Their chosen comment is brief—mentioning living nearby and pointing towards regional attitudes without much emotional support. Rejected was "Un bon gros fdp en somme." Hmm, maybe the rejection indicates preferring shorter responses instead of longer French phrases. Also, the chosen comment shows understanding via location-related insight rather than direct empathy.

Moving to Post 2: A bisexual guy struggling with dating. Chosen reply offers sympathy ("bestest") briefly, whereas the longer, detailed rejection includes advice and humor. Maybe shorter responses resonate better, avoiding verbose explanations. Or perhaps the user prefers conciseness despite the topic complexity.

Third Post discusses aging positively. Selected reaction is simple gratitude, rejected uses dark humor ("immortal"). Choosing the simple thank-you suggests preferring straightforward replies over sarcastic takes.

Fourth Post involves workplace discrimination. User picked a brief complaint asking for legal protections, rejecting a practical legal guide filled with steps. Again, shorter, emotional statements preferred over informative/detailed guidance.

Looking at patterns:

- **Conciseness**: All chosen replies are shorter, direct messages. Reject longer, helpful options.

- **Empathy Focus**: Chosen responses often show shared experiences (e.g., location note implying regional bias) or brief solidarity, not offering solutions.

- **Avoidance of Sarcasm/Judgment**: Some rejected answers use dark humor or harsh tones.

- Short sentences preferred over lengthy ones except when necessary.

Personality-wise, maybe high agreeableness (preferring compassionate responses), low openness to complex advice. Prefers brevity over detail, dislikes long form suggestions. Emotionally, seeks empathetic validation quickly rather than getting bogged down in practicalities.

So summarizing: Preferences lean towards succinct, empathetic communication avoiding unnecessary elaboration. Prefer concise replies highlighting common issues rather than deep analysis. Dislikes lengthy guidance or sarcasm, gravitates toward brief affirmations.

</think>

<answer>
User favors **concise, emotionally attuned responses** over elaborate or directive ones. Key preferences include:

1. **Brevity**: Chooses shorter, punchier remarks (e.g., "I live 20 mins...", "Thanks!") over lengthier explanations.

2. **Empathy Over Detail**: Prioritizes relatable context (regional bias, shared struggles) rather than offering structured solutions.

3. **Avoidance of Sarcasm/Judgment**: Rejected responses contained irony or negativity (dark humor, legal jargon-heavy advice).

4. **Need for Quick Validation**: Prefers brief affirmations ("Thanks", "Sorry") over prolonged discussions.

These preferences suggest a tendency for **concise communication**, valuing immediacy and vulnerability, while shunning complexity or prolonged guidance.
</answer>

---

**Prompt for Updating the 4 Behavior Signals under the Streaming Setting**

A conversation between User and Assistant. The User asks a question, and the Assistant solves it. The Assistant first thinks about the reasoning process in the mind and then provides the User with the answer. The reasoning process is enclosed within <think> </think> and answer is enclosed within <answer> </answer> tags, respectively, i.e., <think> reasoning process here </think> <answer> answer here </answer>. User: You must put your answer inside <answer> </answer> tags, i.e., <answer> answer here </answer>.

This is the problem: Summarize user preferences based on their behavior. If past preferences are provided, adjust the preferences by combining past preferences with those reflected in current behavior, removing conflicting parts, and integrating new insights. If no past preferences are provided, derive the final preferences solely from user behavior.

**Past Preferences:**

User favors **concise, emotionally attuned responses** over elaborate or directive ones. Key preferences include:

1. **Brevity**: Chooses shorter, punchier remarks (e.g., "I live 20 mins...", "Thanks!") over lengthier explanations.

2. **Empathy Over Detail**: Prioritizes relatable context (regional bias, shared struggles) rather than offering structured solutions.

3. **Avoidance of Sarcasm/Judgment**: Rejected responses contained irony or negativity (dark humor, legal jargon-heavy advice).

4. **Need for Quick Validation**: Prefers brief affirmations ("Thanks", "Sorry") over prolonged discussions.

These preferences suggest a tendency for **concise communication**, valuing immediacy and vulnerability, while shunning complexity or prolonged guidance.

**This person has chosen or rejected comments on some posts:**

1. **Post:** This is just a vent and, of course, it's not directed to my lovely spouse, since she's supportive and great. Still, it bothers me so much that people have such a hard time respecting my pronouns, yet as soon as she came out, everybody started using her name and pronouns correctly (she's MtF). We're both at the same stage in our transitions (pre-everything) and when she came out, I could see the immediate change in her friends. She only found out later in life that she's trans, while I've been struggling with it my whole life. It's as if just because I was AFAB, my transition somehow has to be taken less seriously. I don't know, maybe I'm exaggerating. I'm just pretty bummed, mates. My pronouns hardly get respected, yet everyone respects my wife's.

**Chosen:** Dealing with the exact same, actually. Literally the same. Down to the last letter. Sorry brother.

**Rejected:** I always seek solace in the fact that those who misgender me are going to look absolutely dumb one day.

2. **Post:** she said she is gonna support me if i think im trans (in her words) for much longer, but said im not allowed a haircut because i will hate myself more apparently. how the fuck do i even react to this? she makes these random suggestions based off of her own knowledge rather than fact and wont let me correct her. she said i never shower signs of when i was younger, but i did, she just never noticed it. i dont know what to do. i came out to my mum and she said im going through a phase.

**Chosen:** mine said to my father "you know how SHE is, SHE was just venting" when I came out to them. you know what's good for you, nobody else does. she just needs time to understand it. hold on bro

**Rejected:** Go get your hair cut anyway. Any friends can bring you?

3. **Post:** So I am bi. Have always been more attracted to women than men. Sometimes I wonder am I gay? Who knows. I'll figure it out. Some background about me: I haven't been able to meet a woman to date long term. I have mostly only hooked up with women one night stand style, or friends with benefits. I can easily meet men, but I always meet women simultaneously. Any guy I am with has to be 100% ok with that. The women I have really really crushed on or fell in love with either moved away or just dropped off the face of the earth and stopped answering my calls. (the reason for this back story is because I think I would have figured out an answer to my question if I had had a long term gf.) I met a woman the a few weeks ago and we hit it off. I didn't want to have sex with her right away, even though I could tell she was hinting toward it. I wanted to have a chance to explain to her what I enjoy and have a mutually pleasurable experience. I told her I had to end the night and that I would like to have dinner with her in a few days. We had dinner, drinks, more drinks, and by this time we were both pretty tipsy and I felt comfortable enough to flirt and tell her that if we ever got together that I need a lot of warm up. I thought I was pretty clear in explaining that I have a VERY sensitive clit, but we got back to my place and I kissed her, and and then all of a sudden, she is violently mashing my vulva with her face and aggressively fingering me. I just kind of ended it because I said I had too much to drink. I wasn't enjoying it at all and I felt a little violated after I specifically told her the parameters of my body. So. Here's the thing! This happens to me. A lot. I don't really understand why. Even with a disclaimer of "I need to warm up with lots of touching and kissing and teasing and I need a feather light touch to get me going." Do I have the most sensitive clit ever had on a woman's body? Is it more common to just mash the vulva around and suck on a clit with full force? Do most women enjoy just one kiss and then straight to deep aggressive fingering?

**Chosen:** Not alone. I'm not fond of direct touching on my clit, I'd rather someone rub it through the folds around it. I've had girlfriends who have consistently ignored this, went straight for the clit every time, and then got pissed when I just squirm around uncomfortable while they're giving me, in their opinion, the best sex ever.

**Rejected:** No, you sound totally normal. Your problem isn't sensitivity, it's that you're not being assertive enough. Instead of saying "oh, I had too much to drink let's stop", you need to tell your partner "hey, that's way too fast/too hard. You need to slow down." Don't even take your panties off if you're not ready for that, just tell her "not yet, kiss me here", etc. If she gets mad or refuses to listen, end it there (but don't lie about the reason).

4. **Post:** A little rant. My dad said that to me recently. I came out to him like 2 years ago. He said he was ok with me being gay after a week of not talking to me after I came out to him. What I'm wondering is how he'd feel when I'd get a girlfriend (I never had dated yet. I just turned 19 in September). Would having a girlfriend be too shove-y for him? I don't know. I wish he could just tell me straight up that he

isnt́ really that accepting. He is a conservative so I guess I shouldnt́ be surprised. A lot of people since lgbt started becoming more accepting have said stuff like that a lot. What are youŕe experiences when it comes to people who say these types of phrases and how do you feel about it? [Rant] "Iḿ ok with gay people ... Just dont́ shove it in my face"

**Chosen:** When I came out, everyone but one sister "accepted me." Now, every single family member has blocked me on Facebook, except my two gay uncles. One sister-in-law has completely cut me out to the point of not allowing me near their children, only exception being family events like my momś recent birthday. Saddest part was when my little 4-year-old nephew asked me, "why havent́ I seen you in so long???" I had to change the subject while fighting back tears. Thank god for my gay uncles. Iḿ going to make a point to visit them ASAP. *hugs* for everyone going through similar rough patches with toxic people.

**Rejected:** I hate people like that. Its just a bullshit double standard. Heterosexuals *constantly* do the same things that, when done by gay couples, are "shoving it in the face," and nobody even blinks. Can you imagine the shitstorm that would go down if even half of LGB people started telling straight people to stop shoving it in our faces whenever they did things like hold hands in public?

Assistant: \<think\>

---

**Preference Generated by ALIGNXPLORE-7B after Updating the 4 Behavior Signals under the Streaming Setting**

Okay, so I need to summarize the user's preferences using the given framework. Let's see...
First, looking at the past preferences:
They prefer concise, empathetic responses avoiding sarcasm or judgment. They value quick validation without long explanations. So, their communication style leans towards brief and heartfelt messages rather than lengthy ones.
Looking at the examples provided:
In Post 1, the user chose a response that empathizes ("dealing with the same..."), which fits brevity and shared struggle. Rejected the sarcastic comment which uses dark humor.
For Post 2, the selected reply offers comfort and practical advice ("nobody needs to rush understanding") whereas the rejected option suggests a humorous question, perhaps seen as dismissive.
On Post 3, the preferred reaction validates the experience (mentioning discomfort with direct contact) without giving specific advice. The rejected suggestion pushes active correction, which might feel abrupt.
Lastly, in Post 4, the accepted message shares personal relatable pain succinctly, whereas the rejected one introduces a broader critique which could extend beyond the immediate situation.
So overall, the user prefers responses that acknowledge shared feelings, validate their situation, and avoid pushing solutions or jokes. They avoid anything sarcastic or overly directive. Their needs align with concise empathy, avoiding negativity, and seeking immediate connection.
\<think\>

\<answer\>
The user consistently favors **concise, emotionally resonant statements** that validate shared experiences and foster vulnerability, shunning humor, sarcasm, or prescriptive advice. Key traits observed include:

1. **Brevity**: Selects brief, relatable anecdotes (e.g., "Same situation"). Rejects verbose/detailed suggestions.

2. **Empathy-driven validation**: Chooses responses acknowledging pain (e.g., "Saddest part was when my little 4-year-old nephew...") instead of analytical critiques.

3. **Avoidance of judgment/negative framing**: Dismisses sarcastic remarks ("dumb"), ironic questions, or framing problems as moral failures.

4. **Value for brevity**: Prefers unadorned affirmations ("Thanks!") over elaborated guidance.

These choices reflect a preference for **empathetic, unstructured solidarity** over structured problem-solving, emphasizing immediacy and emotional authenticity in communication.
</answer>

## C   DATA FORMAT AND PROMPT

**Evaluation Prompt**

Determine which response the user prefers based on the user's preferences. Please output your selection below in a json format by filling in the placeholders in []:

```
{"selection": "[Response A / Response B]"}
```

**Prompt**
It is just recently that I realized I have no idea what the job market for chemists looks like, although I am a chemistry student. Could someone shed some light on that? Also, which of those bring in the most money?
Chemistry jobs?

**Preference**
**This person's persona is:**
This person likes science, knowledge. An aspiring chemistry professional who values optimism and factual information, preferring to gather insights from collective experiences rather than individual anecdotes. This person is proactive in seeking career guidance and appreciates straightforward, encouraging responses.

**Response A**
I just had a job interview with a private analytical company a week ago. They were small and privately owned. You could have considered it an entry level position. They weren't expecting anyone to be FULLY trained in their analytical methods, which were not very common. I found out after I DIDN'T get the job that I had to compete with two Ph.D.s and a ton of other people. I'm fresh out of undergrad. Twas my first job interview and I am glad I had the experience of getting that for the next few job interview experiences I have possible coming up, but wow, I was getting screened for that job while two other doctorate fellows were also totally into it.

**Response B**
I think the people in this thread are a bit pessimistic. Someone did a salary thread a few weeks ago and it didn't look bad at all. One thing a prof mentioned to me is that companies pretty much will not hire someone without lab experience as other people have mentioned.

```
{"selection": "Response B"}
```

**Prompt**
{prompt}

**Preference**
{persona}

**Response A**
{responseA}

**Response B**
{responseB}

> **Prompt for Generating Reasoning Chains and Preference Descriptions under Base Setting**
>
> Generate the user's preference based on their historical behavior.
> The following aspects can be referred to when analyzing user preferences.
> {key preference dimensions}
>
> **This person has chosen or rejected comments on some posts:**
> {implicit preference signals}

> **Prompt for Generating Reasoning Chains and Preference Descriptions under Streaming Setting**
>
> Summarize user preferences based on their behavior. If past preferences are provided, adjust the preferences by combining past preferences with those reflected in current behavior, removing conflicting parts, and integrating new insights. If no past preferences are provided, derive the final preferences solely from user behavior.
> The following aspects can be referred to when analyzing user preferences.
> {key preference dimensions}
>
> **Past Preferences:**
> {past preferences}
>
> **This person has chosen or rejected comments on some posts:**
> {implicit preference signals}

# D    LIMITATIONS

Due to the lack of a real LLM-user interaction test platform, we were unable to validate the model's reasoning performance in a real-world environment. Once such a testbed becomes available, we will evaluate our model's performance on it. This paper primarily focuses on the scenario of preference inference, ensuring that the historical preferences in the test set are consistent with the test pairs. Future work could extend to scenarios where user preferences change dynamically over time, requiring the model to adjust preferences based on the user's recent behaviors during inference.

