# OpenReview forum: "Extended Inductive Reasoning for Personalized Preference Inference from Behavioral Signals"
_ICLR.cc/2026/Conference — Submitted to ICLR 2026_

### Official Review · Reviewer_wURy · 2025-10-24

**Soundness:** 3
**Presentation:** 2
**Contribution:** 2
**Rating:** 4
**Confidence:** 3

**Summary:**

This paper addresses the challenge of inferring personalised preferences via extended inductive reasoning. To this end, the authors propose AlignXplore, which is a model that uses reasoning chains to systematically infer preferences from signals in the user's interaction history. They do this by composing multiple methods together, such as (i) creating a synthetic dataset on which they do SFT, and (ii) a GRPO-based reinforcement learning component. To empirically validate their model, they test it on their own validation dataset (AlignX) and on P-Soups, where their methods seem to show promising results.

**Strengths:**

The paper addresses an interesting problem, and the authors conduct extensive empirical validation of their hypotheses. The results of these experiments seem promising, which speaks for their proposed methods.

**Weaknesses:**

While the experiments are extensive in this paper, and the appendix provides a lot of details, I believe there are specific ways this paper can be improved, and I look forward to a discussion with the authors. My main concerns are the following:

- **No statistical significance**: All the experiments do not provide any error bars (e.g. confidence intervals, standard errors, standard deviations), to gauge if the results are actually statistically significant. Table one mentions a t-test and significance for some of the reported results, but to me, it is unclear against which method they compare the t-test. I would therefore strongly encourage the authors to either run multiple runs (optimally, though I understand this is time- and compute-consuming) or, at a minimum, run a bootstrapped confidence interval on the test set to examine 95%-CIs.

- **Composition of Many Methods**: The proposed method by the authors (at least to me) seems more like a composition of various methods, and the claimed results are only achieved if all the techniques are combined. Now, per se, I believe that there is not much wrong with combining existing methods. However, in this case, I find it difficult for the reader to follow and understand which method actually contributes how much. While Figure 1 provides a good overview of what is happening, it feels overfilled with information, and I am trying to understand the authors' contribution.

**Questions:**

I have some questions:

- In paragraph 4.2 Main Results, the authors mention in point (5) that RL is the dominant training stage, and that there is an ablation for it. Where exactly do the authors take this insight from? Is it from Figure 4? A more extensive discussion that justifies the points in that section would be helpful.

- What point do the authors wish to make in Figure 5? While the figure looks pretty, to me, it is not particularly informative, and I would move it to the Appendix.

---

> ### Author Response · Authors · 2025-11-21
> **Response to Reviewer wURy (Part 1)**
>
> > W1: No statistical significance: All the experiments do not provide any error bars (e.g. confidence intervals, standard errors, standard deviations), to gauge if the results are actually statistically significant. Table one mentions a t-test and significance for some of the reported results, but to me, it is unclear against which method they compare the t-test.
>
> We completely agree that providing robust statistical significance is essential for validating our results:
>
> 1. **Clarification of the Original T-Test:** To first clarify the t-test in the original Table 1, the asterisk (`*`) on a baseline's result indicated that our final model, AlignXplore-7B (Streaming, T=4), performed statistically significantly better than that specific baseline in a paired t-test (p < 0.001). We acknowledge this was not clearly explained in the caption and was an inadequate way to show significance.
> 2. **New Multi-Run Experiments for Statistical Validation:** More importantly, to directly address your concern and follow your excellent suggestion, we have now conducted multiple runs (10 runs with different random seeds) for our main model and key baselines. The results, presented as mean ± standard deviation, are shown below:
>
> |Method|AlignX$_{\rm test}$|P-Soups (Informativeness)|P-Soups (Style)|P-Soups (Expertise)|
> |-|-|-|-|-
> |Qwen2.5-7B-Instruct (4 pairs)|0.579 ± 0.005|0.517 ± 0.010|0.610 ± 0.030|0.665 ± 0.010
> |DS-R1-Distill-Qwen (Streaming, 4 pairs)|0.556 ± 0.006|0.511 ± 0.013|0.500 ± 0.022|0.583 ± 0.018
> |**AlignXplore-7B (Streaming, 4 pairs)**|**0.721 ± 0.008**|**0.599 ± 0.010**|**0.814 ± 0.015**|**0.710 ± 0.024**
>
> As the table clearly demonstrates, the performance gains of AlignXplore are not only substantial but also consistent across runs. The mean accuracy of our model is significantly higher than the baselines on all metrics, with the improvements far exceeding the observed standard deviations. This confirms that our results are statistically significant and robust. We will update Table 1 to include the mean and standard deviation for these key comparisons and add a discussion on the statistical significance in the revision.
>
> > W2: Composition of Many Methods: I find it difficult for the reader to follow and understand which method actually contributes how much.
>
> We sincerely thank the reviewer for acknowledging the effectiveness of our approach. We understand the concern about discerning the individual contributions and the density of Figure 1. We would like to clarify our contribution and how we will improve the paper's presentation.
>
> 1. We respectfully clarify that AlignXplore is not an ad-hoc composition, but an integrated framework where each core component is mutually reinforcing and essential for achieving effective and efficient personalization:
> 	- The **Explicit Preference Inference Paradigm** is our foundational design choice, necessary to overcome the opaqueness and inefficiency of implicit methods. The value of this explicit approach is demonstrated in our results (Table 3), where our inferred preference summary significantly outperforms baselines that use raw signals, validating this core paradigm.
> 	- The **Efficient Streaming Mechanism** is a direct and powerful consequence of this paradigm, and it is essential for practical, real-world applications where user history grows over time. This is not an independent feature but the component that delivers the *efficiency* of our approach, as quantified by the reduced computational cost and latency shown in Figure 3.
> 	- The **Two-Stage Training Framework (SFT+RL)**, while a known pattern, required significant, non-trivial engineering to apply successfully to our inductive inference task. Its implementation is a core part of our contribution, involving several key insights in guided data synthesis for a strong cold-start (§3.2, Lines 200-203), the engineering of a self-correcting, error-tolerant streaming mechanism (§3.2, Lines 206-207), and nuanced reward design for effective training (§4.6, Finding 1).
>
> 	In essence, our paradigm defines an effective goal, the training framework makes it achievable, and the streaming mechanism makes it efficient in practice. We will revise the introduction to better articulate this synergy.
>
> 2. We agree that the original Figure 1 was dense and tried to convey too much information. To address this and better highlight our contribution, we will split Figure 1 into two separate, more focused figures.
> 	- A new **Figure 1 (Task Overview)** will be simplified to cleanly illustrate the core task and our streaming inference mechanism.
> 	- A new, larger **Figure 2 (Training Pipeline)** will be dedicated to clearly laying out the two-stage training process, making the flow and synergy of SFT and RL much easier to follow.
>
> We are confident that these clarifications, supported by our ablation results and the redesigned figures, will make the unique contribution of our synergistic framework more straightforward.

---

> > ### Author Response · Authors · 2025-11-21
> > **Response to Reviewer wURy (Part 2)**
> >
> > > Q1: In paragraph 4.2 Main Results, the authors mention in point (5) that RL is the dominant training stage, and that there is an ablation for it. Where exactly do the authors take this insight from? Is it from Figure 4? A more extensive discussion that justifies the points in that section would be helpful.
> >
> > Our claim is based on the **ablation study in Table 1**. For clarity, we have extracted the relevant results from Table 1 (all in the `base` setting) into the table below:
> >
> > |Method|AlignX$_{\rm test}$|P-Soups (Informativeness)|P-Soups (Style)|P-Soups (Expertise)|
> > |-|-|-|-|-
> > |AlignXplore-7B|**65.33** (0.00)|**54.32** (0.00)|**69.67** (0.00)|**63.83** (0.00)
> > |AlignXplore-7B w/o RL|61.80 (-3.53)|52.82 (-1.50)|54.00 (-15.67)|59.83 (-4.00)
> > |AlignXplore-7B w/o Cold-start|62.80 (-2.53)|56.64 (+2.32)|64.83 (-4.84)|59.50 (-4.33)
> >
> > Values in parentheses indicate the change relative to the full AlignXplore-7B model. From this comparison, we draw two key conclusions:
> > - On AlignX$_{\rm test}$, removing the RL stage (w/o RL) causes a significantly larger performance drop (-3.53 points) than removing the cold-start SFT stage (-2.53 points).
> > - This effect is even more pronounced on the out-of-domain P-Soups dataset, especially for the `Style` subset, where the RL stage appears crucial for learning the nuanced aspects of high-quality preference generation.
> >
> > While the primary purpose of Figure 4 is to analyze RL training dynamics instead of final performance, it does provide some support for our claim: The final reward level of the `w/o Cold-start` run (the orange curve) at step $\approx$ 200 is visibly higher than the initial reward of the full `AlignXplore-7B` run (the blue curve) at step 0, i.e., the performance of the Cold-start-only model. However, this is not a rigorous comparison, as the training data at different steps varies. Our primary evidence remains the direct ablation in Table 1.
> >
> > In the revision, we will add a direct reference (e.g., `(see Table 1, w/o RL)`) in the text and create a dedicated **"Ablation Studies" subsection** to make this crystal clear.
> >
> > > Q2: What point do the authors wish to make in Figure 5? While the figure looks pretty, to me, it is not particularly informative, and I would move it to the Appendix.
> >
> > The purpose of Figure 5 is to provide crucial **qualitative evidence** for our paper's central claim: that our two-stage training progressively refines descriptions from raw observation into actionable synthesis. The figure visually demonstrates a clear progression:
> >
> > - **The baseline LLM** generates descriptive, non-synthesizing terms (e.g., "historical," "situation").
> > - **Our SFT-only model (`w/o RL`)** begins to identify preference *dimensions* (e.g., "communication style") but lacks synthesizing language.
> > - **Crucially, our full model** is the only one that **synthesizes** these dimensions with **actionable verbs** (e.g., "prioritize," "avoid," "leans toward").
> >
> > This visual evolution from observation to actionable hypothesis is the qualitative demonstration of our method's mechanism, and intuitively showcases the common patterns of the inductive reasoning process. For this reason, we believe it is essential for the main paper. We will substantially revise the figure caption and in-text discussion to explicitly guide the reader through this analysis, ensuring its role as key qualitative evidence is immediately clear.

---

> > > ### Comment · Reviewer_wURy · 2025-11-25
> > >
> > > Thank you very much for responding. I will address all the answers below:
> > >
> > > *A1) errorbars*
> > > Thank you for providing the error bars for the three combinations you used. It seems like the results are stable there, looking at the standard deviation. It would be great if all the experiments included these to determine the method's significance.
> > >
> > > *A2) clarification of the method*
> > > Thank you for clarifying the method. I acknowledge that. While I appreciate your promise to update the figure and the manuscript, I have not seen any changes yet in the manuscript. From my understanding, the authors are allowed to upload a new version of the manuscript for the reviews.
> > >
> > > *A3)* Thanks for the follow-up experiments on the ablation of the various parts. Any chance of introducing the error bars here again to determine how significant each method is to improve, or if this is just stochasticity from the data and optimiser?
> > >
> > > *A4) Qualitative Assessment*
> > > Thank you for pointing out the reason for Figure 5. However, I still find this pretty difficult to read in a world cloud format like this. I would recommend, e.g., making a bar plot of the 15 most important concepts and showing how they are ranked differently, as the size and colour (to me) do not seem particularly indicative.
> > > Nonetheless, I leave this to the authors' judgment.
> > >
> > > I would be willing to raise my score if the ablation of A3) shows significant results of the method, and the changes of clarity are made in the updated manuscript.

---

> > > > ### Author Response · Authors · 2025-11-26
> > > > **Response to Reviewer wURy**
> > > >
> > > > We sincerely thank you for your continued engagement and constructive feedback. We appreciate your willingness to reconsider your score based on these clarifications and additional experiments.
> > > >
> > > > > Q1: Error Bars for All Experiments
> > > >
> > > > We fully agree that providing error bars for all experiments would further solidify the significance of our findings. However, due to the substantial computational cost required to repeat every reported experiment multiple times (10$\times$), it is challenging to complete this for all baselines within the limited rebuttal window. We will conduct these additional runs and include the full statistical analysis in the Appendix of the final revision to further strengthen the validity of our results.
> > > >
> > > > > Q2: Manuscript Updates
> > > >
> > > > We apologize for the delay in visibility. We have now uploaded the revised manuscript. As promised, we have rewritten the Introduction and Methodology (changes marked in red) to clearly articulate the synergy of our framework. Furthermore, we have split the original dense Figure 1 into two separate figures: a new Figure 1 illustrating the Preference Inference Task and a new Figure 2 detailing the Two-Stage Training Process. These changes are now available in the updated manuscript.
> > > >
> > > > > Q3: Statistical Significance of Ablation Studies
> > > >
> > > > Thank you for this crucial suggestion. To verify that our ablation findings are not due to stochasticity, we conducted 10 independent runs with different random seeds for the ablation settings. The results (Mean ± Standard Deviation) are reported below:
> > > >
> > > > |Method|AlignX$_{\rm test}$|P-Soups (Informativeness)|P-Soups (Style)|P-Soups (Expertise)|
> > > > |-|-|-|-|-
> > > > |**AlignXplore-7B**|**0.651 ± 0.006**|**0.553 ± 0.019**|**0.685 ± 0.027**|**0.653 ± 0.019**
> > > > |AlignXplore-7B w/o RL|0.602 ± 0.005|0.527 ± 0.013|0.580 ± 0.022|0.576 ± 0.018
> > > > |AlignXplore-7B w/o Cold-start|0.628 ± 0.007|0.561 ± 0.015|0.600 ± 0.033|0.611 ± 0.018
> > > >
> > > > These results demonstrate that AlignXplore-7B w/o Cold-start consistently and significantly outperforms AlignXplore-7B w/o RL (e.g., on AlignX$_{\rm test}$, **0.628 vs. 0.602** with low variance). This statistically confirms our claim that while both stages matter, the RL stage is the dominant factor in driving performance improvements.
> > > >
> > > > > Q4: Qualitative Assessment (Figure 5)
> > > >
> > > > We appreciate your feedback regarding the readability of the word clouds. Your suggestion to use a bar plot to rank the most important concepts is insightful. We will carefully consider generating these plots and incorporating them into the Appendix of the final revision to further enhance the clarity of our qualitative analysis.

---

> > > > > ### Comment · Reviewer_wURy · 2025-11-26
> > > > >
> > > > > *A1) errorbars*
> > > > > Another way could be to bootstrap the test set metric and calculate the confidence interval. That would not require you running the experiments multiple times.
> > > > >
> > > > >
> > > > > Thank you, my concerns have been addressed (or are promised to be addressed), and I raised my score accordingly.

---

> > > > > > ### Author Response · Authors · 2025-11-26
> > > > > > **Response to Reviewer wURy**
> > > > > >
> > > > > > We sincerely thank you for raising the score and for the insightful suggestion regarding bootstrapping.

---

### Official Review · Reviewer_VE99 · 2025-10-31

**Soundness:** 3
**Presentation:** 2
**Contribution:** 2
**Rating:** 4
**Confidence:** 3

**Summary:**

The authors identify that inductive reasoning, the ability to infer general principles from incomplete evidence, is underexplored in LLMs, and they investigate it through the task of preference inference. They propose AlignXplore, an LLM designed to infer explicit user preferences from implicit behavioral signals. Taking synthetic data generated by a teacher model QwQ32B, the LLM is fine-tuned using SFT + GRPO. To handle evolving user preference, the authors introduce a streaming inference mechanism that refines inferred preferences based on new signals without reprocessing the entire history interaction. Experiments show an average performance improvement of 15.49% on both in-domain and out-of-domain datasets, with additional analyses indicating strong generalization, robustness, and efficiency.

**Strengths:**

1. Streaming preference inference is an interesting direction and the proposed method directly targets this goal. The approach allows the model to incrementally refine inferred preferences as new signals come in, without reprocessing the entire user history, improving efficiency significantly.
2. The experiments are systematic and comprehensive, covering both in-domain and out-of-domain datasets while evaluating generalization, robustness, and efficiency. The authors run various ablation studies to provide insights into the contribution of each component, which supports the validity of the overall design.

**Weaknesses:**

1. Although the paper claims to explore extended reasoning for preference inference and lists several advanced reasoning mechanisms in related work, the actual experiments are limited to basic reasoning chains. As a result, the interaction between preference inference and extended reasoning may not be deeply explored, offering limited novelty.
2. The method follows a standard SFT + GRPO pipeline using synthetic data generated from QwQ-32B. Given similar performance compared to off-the-shelf QwQ-32 (65.33 vs 65.7), it may effectively distill the teacher model's behaviors, but the algorithm/technical implementation may not provide sufficient novelty.
3. Preference descriptions are fully open-ended, lacking constraints on factors like granularity or structure, which may potentially lead to inconsistent preference descriptions and unstable downstream performance. In addition, there's no human study to manually check the quality of preference description generated from QwQ-32B.

**Questions:**

1. In section 3.2, my understanding is each signal or post e_i may be associated with a reasoning chain r_i. Could you clarify what you meant by d_i? For a set of signals, do you assume a separate preference description for each or a consolidated preference description?
2. In section 4.5, regarding the accuracy drop, could this be partially attributed to a train-test mismatch? i.e. the model is trained on sequences of four signals (T=4) but tested on longer sequence (N=16)?

---

> ### Author Response · Authors · 2025-11-21
> **Response to Reviewer VE99 (Part 1)**
>
> > W1: Although the paper claims to explore extended reasoning for preference inference and lists several advanced reasoning mechanisms in related work, the actual experiments are limited to basic reasoning chains.
>
> We thank the reviewer for this insightful question. Our response is two-fold:
>
> 1. **We define "Extended Reasoning" as Deliberate Thinking Time:** The "extension" refers to the *length and deliberateness* of the thought process, not its algorithmic complexity.  This concept aligns with the paradigm in models like OpenAI o1 [1] and Anthropic's "extended thinking" [2] that spend more computational effort before responding. In our work, this involves a long-horizon path from behavioral signals to a final preference, as opposed to having the model directly produce a preference from those signals.
>
> 2. **Our Novelty: Pioneering This Paradigm to Inductive Personalization:** Our primary novelty is not in inventing a new reasoning algorithm, but in pioneering the use of this "extended thinking" paradigm for the challenging, inductive domain of personalized preference inference.
>
> This reframes your valid observation about "basic reasoning chains" not as a limitation, but as a deliberate experimental design intended to test a core hypothesis: that this paradigm is essential for effective inductive personalization. Our evaluation in Table 1 was structured precisely to validate this by systematically comparing:
> - `Extended Reasoning = N/A`: Methods including both non-reasoning baselines and specialized systems with different reasoning structures (e.g., LMInductReason).
> - `Extended Reasoning = ×`: Strong LLMs prompted for **short, standard reasoning**.
> - `Extended Reasoning = √`: Our approach and other LLMs guided by **long, extended reasoning**.
>
> The results show that the (√) category consistently and significantly outperforms the others. In the revised manuscript, we have added Footnote 1 on Page 2 and updated the Introduction (Paragraph 3) to clearly articulate this contribution.
>
> **References**
>
> [1] OpenAI. Introducing OpenAI o1. 2024
>
> [2] Anthropic. Claude’s extended thinking. 2025.
>
> > W2: The method follows a standard SFT + GRPO pipeline using synthetic data generated from QwQ-32B. Given similar performance compared to off-the-shelf QwQ-32 (65.33 vs 65.7), it may effectively distill the teacher model's behaviors, but the algorithm/technical implementation may not provide sufficient novelty.
>
> While the comparison to the teacher model is a valid starting point, we believe it leads to an incomplete picture. Our novelty is not in the SFT+GRPO pipeline itself, but in **what we achieve with it**:
>
> 1. **Our Core Innovation: Pioneering a New Paradigm for Personalization.** Our primary contribution is pioneering the "extended thinking" paradigm for the challenging, inductive domain of personalized preference inference. This differs fundamentally from self-contained deductive tasks (e.g., math, code). For the unique challenge of evolving user histories in personalization, we introduced a key component of our novelty: an efficient streaming mechanism. This is made possible by our inductive synthesis of a preference summary, which creates a compact, updatable state that does not exist in typical deductive reasoning workflows.
> 2. **Our Results: Superior Performance and Efficiency, Not Simple Distillation.** The success of our approach is evident when comparing the appropriate models in their intended settings.
> 	- **Establishing the Baseline:** The reviewer rightly points to our `base` model's performance (65.33) being similar to the teacher's (65.7). We view this as a successful first step: it validates that we can distill the teacher's static, single-pass reasoning ability into a smaller model.
> 	- **Demonstrating the New Capability:** However, our ultimate goal was never to simply replicate this static performance. The `base` setting suffers from known limitations (performance collapse with long history), which is precisely why we developed the `streaming` setting. Our final model, **AlignXplore-7B (Streaming), achieves 71.47%**. This result is not a simple distillation; it significantly surpasses the static teacher model (QwQ-32B) at 65.70. More strikingly, it also outperforms even a much larger, state-of-the-art reasoning model, **DeepSeek-R1-671B**, in its streaming configuration (67.70).
> 	- **Efficiency:** As Figure 4 demonstrates, our state-of-the-art performance is achieved with **stable, low inference latency**, while the non-streaming "Base Setting" becomes impractically slow.
>
> Therefore, the combination of superior streaming performance and high efficiency demonstrates that our work is far more than distillation. We have engineered a specialized model that learns a **new, practical capability** that the original teacher model does not possess. We have updated the Introduction to make this crucial distinction and contribution clearer.

---

> ### Author Response · Authors · 2025-11-21
> **Response to Reviewer VE99 (Part 2)**
>
> > W3: Preference descriptions are fully open-ended, lacking constraints on factors like granularity or structure, which may potentially lead to inconsistent preference descriptions and unstable downstream performance. In addition, there's no human study to manually check the quality of preference description generated from QwQ-32B.
>
> We conducted targeted studies to address the concerns regarding the potential for inconsistency, performance instability, and the lack of human validation.
>
> **1. On Preference Description Quality and Inconsistency**
>
> We directly evaluated our open-ended descriptions with human experts.
> - **Method:** We randomly sampled 200 preference descriptions generated by QwQ-32B. Three human evaluators scored each on a 1-8 scale across key criteria, including:
> 	(1) Groundedness: Is the description supported by the provided user signals?
> 	(2) Comprehensiveness: Does it capture the key aspects of the user's preference?
> 	(3) Consistency:  Is the description internally logical?
> 	(4) Clarity: Is the description easy to understand?
> - **Result:** The descriptions achieved a high average quality score of **6.76 out of 8**, with an inter-annotator agreement of Fleiss' $\kappa = 0.54$. This directly confirms that our generation process produces high-quality, reliable, and consistent preference descriptions, mitigating the concern of uncontrolled open-ended generation.
>
> **2. On Unstable Downstream Performance**
> - **Method:** We repeated our main experiments across 10 random seeds for our model.
> - **Result:** Our AlignXplore-7B (Streaming) model achieved a final accuracy of **72.1% ± 0.8%**. The results show extremely low variance, confirming the stability of our end-to-end pipeline.
> - **Conclusion:** This minimal standard deviation provides strong empirical evidence that our method is robust and that the open-ended nature of the intermediate descriptions does not lead to performance instability.
>
> **3. On the Lack of Human Validation**
>
> We evaluated the alignment between our automated filtering process and human judgment.
>
> - **Method:** We randomly sampled 200 unfiltered preference descriptions generated by QwQ-32B. For each, we compared the preference judgment made by our automated LLM-based filter to the judgment of three human evaluators performing the same task.
> - **Result:** We found a strong agreement rate of **80%** between the automated filter and human decisions, with an inter-annotator agreement of Fleiss' $\kappa = 0.53$. This high alignment demonstrates that even with open-ended preference descriptions, our downstream selection process is stable, reliable, and closely mimics human reasoning. It is not an arbitrary or unstable process.
>
> These targeted studies—a direct human evaluation of description quality and robust multi-seed experiments—confirm that our method is reliable, consistent, and stable. We will add a new appendix detailing these validation studies to the revised manuscript.
>
> > Q1: In section 3.2, my understanding is each signal or post e_i may be associated with a reasoning chain r_i. Could you clarify what you meant by d_i? For a set of signals, do you assume a separate preference description for each or a consolidated preference description?
>
> Our model's objective is to generate **a single, consolidated preference description ($d$)** that synthesizes the entire set of a user's signals $\mathcal{E} = \{e_1, e_2, \dots, e_T\}$. The goal is to infer a holistic user profile, not to describe each signal individually. The confusion arose from the overloaded notation $d_i$. This was intended to denote the description of the **$i$-th rollout** during our data synthesis process, not a separate description for signal $e_i$. We have clarified the formulation in Section 3.1 to resolve this.
>
> > Q2: In section 4.5, regarding the accuracy drop, could this be partially attributed to a train-test mismatch? i.e. the model is trained on sequences of four signals (T=4) but tested on longer sequence (N=16)?
>
> Your hypothesis is exactly correct. The performance drop observed in our `base` setting is indeed a direct consequence of the **train-test mismatch** you've identified. The model, having been trained exclusively on sequences of 4 signals, struggles to effectively process and synthesize information from a much longer sequence of 16 signals. This very challenge is the primary motivation for introducing our **`streaming` setting**. By processing signals iteratively, the `streaming` model is specifically designed to handle arbitrarily long sequences, thereby **eliminating this train-test mismatch**. This allows the `streaming` model to maintain stable and robust performance as the number of input signals grows.

---

> ### Author Response · Authors · 2025-11-26
> **Looking forward to your feedback**
>
> Dear Reviewer VE99,
>
> We sincerely appreciate your thoughtful and constructive feedback. In response, we have further clarified the key innovations introduced in our work. To address your concerns regarding the quality of preference descriptions and the stability of the method, we have added additional human evaluation studies as well as variance analyses across multiple runs. We have also provided detailed answers to your two questions to eliminate potential sources of confusion.
>
> We hope these revisions help address your concerns, and we would greatly appreciate it if you could review our responses at your earliest convenience.

---

### Official Review · Reviewer_8Vhm · 2025-10-31

**Soundness:** 2
**Presentation:** 1
**Contribution:** 2
**Rating:** 4
**Confidence:** 3

**Summary:**

This paper proposes a system that infers and incrementally updates explicit preference profiles from the interaction history of a given user.
The model generates a natural-language description that can be used to steer a downstream model's behavior toward an individual user's style.
The approach combines supervised distillation with RL using a learned judge reward, and it supports a streaming update mode for continual personalization.
In the empirical evaluation, a 7B model trained this way outperforms baselines and approaches much larger systems on alignment-style metrics.

**Strengths:**

The paper tackles a very relevant problem of automatic preference personalization for LLMs.
The overall approach is reasonable and interpretable by keeping an explicit, text-based description of the preferences.

**Weaknesses:**

My main concern is about the clarity of presentation - the paper as it is right now is extremely difficult to read and understand, it discusses too many things at once. Some examples:
1. There's a big emphasis on "inductive reasoning", which is technically correct, but seems largely inconsequential to the method and the problem being tackled.
2. Figure 1 is very dense and hard to navigate before having understood the rest of the paper
3. In section 3, there's a lot of redundant notation, like "user U" or "r, d = M(E)" which later turns into "r, d = M(E, \hat{d})". In general mathematical notation seems to be used as a means of making the paper more professional, and not a tool to make the writing precise.
4. Going through Section 3, I was hoping to get a clean, step by step description of the proposed method. The information is all there, but not clear. When a preference scoring function is introduced, it is not described in any way except notation, and later in how it can be instantiated. Even the signature of this function is a bit unclear and has to be inferred from the context - using \cdot as one of the arguments typically indicates that the argument will be filled in later, but in this case it seems to mean that we just omit the description?

While these points might seem like nitpicks, together they significantly

**Questions:**

Please improve the clarity of the paper's writing. The relatively simple and useful ideas are obscured by overstated claims about "extended inductive reasoning" etc.

---

> ### Author Response · Authors · 2025-11-21
> **Response to Reviewer 8Vhm (Part 1)**
>
> We are sincerely grateful for your thorough and constructive feedback. We agree with your assessment and have developed a comprehensive revision plan to address these issues holistically.
>
> **Overview of Planned Revisions**
>
> Our primary goal in this revision is to transform the paper's readability and formal rigor. To that end, we will perform a full-paper polish centered on the following concrete actions:
>
> 1. **Strengthening the Core Narrative (§1):** We will explicitly connect the "inductive reasoning" framework to our method's advantages. We will clarify that this framing defines our task (deriving a general preference from noisy signals) and distinguishes our work from opaque personalization methods (by mandating an **explicit rule**) and from mainstream deductive reasoning research (e.g., math, code). We will then show how this explicit preference is the key that enables our efficient streaming mechanism.
>
> 2. **Improving Visual and Structural Flow (Fig. 1 & §3):** The dense Figure 1 will be split into two separate, more focused figures:
> 	- A new **Figure 1 (Task Overview)** in the Introduction will be simplified to clearly illustrate the core task. It will focus on how our model takes behavioral signals and a historical preference summary to produce an updated, refined preference. The "downstream tasks" component will be streamlined into a single box to avoid early distraction.
> 	- A new, larger **Figure 2 (Training Pipeline)** in the Methodology section will detail the two-stage training process. This will provide the necessary space to clearly illustrate the data flow through both the cold-start SFT and the reinforcement learning stages.
>
> 	This separation ensures that readers can first understand the "what" (the task) before diving into the "how" (the training process), directly resolving the valid concern you raised.
>
> 3. **Overhauling Methodology (§3) for Precision:** We will completely restructure Section 3 for a clean, step-by-step description. This includes:
> 	- **Streamlining Notation:** We will streamline our notation to be more purposeful. Redundant terms like `user U` will be removed. The core inference function will be consistently defined as $r, d = \mathcal{M}(\mathcal{E}, \hat{d})$ with a note that $\hat{d}$ is empty for the initial inference.
> 	- **Unambiguously Defining Functions:** We will rewrite the introduction of the preference scoring function $f_{\mathcal{R}}(y | x, \cdot)$ to be more explicit: (1) Stating its purpose clearly, i.e., scoring a response's alignment with a user's preference; and (2) Defining its signature unambiguously, by clarifying that the `·` argument was intended to represent $d$ (for $\mathcal{R} _ {\text{gen}}$) or $d, y_w, y_l$ (for $\mathcal{R}_{\text{jud}}$). We will replace this ambiguous notation with a clearer, more formal definition.
> 	- **Logical Restructuring:** We will split the current §3.1 into two distinct parts: §3.1 Task Formulation (defining the inputs/outputs) and a new §3.2 Evaluation Framework (defining the reward metrics). This ensures the "what" is clearly established before the "how," creating the clean, sequential description you suggested. The training sections will follow as §3.3 and §3.4.
>
> 4. **Enhancing Experimental Clarity (§4):** We will perform a thorough pass to explicitly define all metrics and baseline implementations, make table captions more self-contained, and split dense tables (like Table 3) to improve readability.
>
> Below, we detail how this comprehensive plan addresses each specific weakness.
>
> **Regarding Weakness 1 (Emphasis on "Inductive Reasoning"):**
>
> We respectfully emphasize that this inductive framework is highly consequential. As we noted in our overview, our revision will make this connection explicit. To elaborate on the key points we will highlight:
> - **Defining and Distinguishing Our Work:** We will clarify how the inductive framing is essential for defining our task, i.e., deriving a general rule (an explicit preference summary) from noisy signals. This distinguishes our work from: (a) **Opaque personalization methods**, which rely on implicit representations, and (b) **Mainstream deductive reasoning** (e.g., math, code), which involves applying known rules rather than discovering new ones.
> - **Enabling Our Core Technical Advantage:** We will show how this framework directly enables our efficient streaming mechanism. This is only possible *because* the inductive process mandates an explicit preference summary, which serves as a compressed, updatable state of the user's history.
>
> By making these connections clear, we will demonstrate that "inductive reasoning" is not just terminology but the theoretical backbone that underpins our method's design and success.

---

> > ### Author Response · Authors · 2025-11-21
> > **Response to Reviewer 8Vhm (Part 2)**
> >
> > **Regarding Weakness 2 (Dense Figure 1):**
> >
> > As outlined in Point 2 of our overview, we will split Figure 1 into two separate figures—a high-level **Task Overview** and a detailed **Training Pipeline**—to dramatically improve clarity and navigability. This allows the reader to first understand the "what" before understanding the "how."
> >
> > **Regarding Weakness 3 & 4 (Clarity and Notation in §3):**
> >
> > We agree entirely that this section lacked the required precision. As summarized in Point 3 of our comprehensive plan, we will overhaul Section 3 to resolve these issues. This includes streamlining all notation, explicitly defining the purpose and signature of all functions (including the scoring function), and restructuring the section for a clean, step-by-step presentation of our method. This directly addresses your concerns about redundant notation and the need for a clearer description.
> >
> > We are confident that this thorough revision will make our methodology and experiments significantly clearer and more precise. We thank you again for your invaluable guidance in improving our work.

---

> > > ### Author Response · Authors · 2025-11-26
> > > **Response to Reviewer 8Vhm (Manuscript Updated)**
> > >
> > > Following up on our previous response, we are writing to confirm that we have now uploaded the revised manuscript. We have thoroughly executed the comprehensive revision plan we outlined (**especially strengthening the inductive reasoning narrative, splitting Figure 1, and restructuring Section 3**) to address your concerns regarding the paper's writing and structure. We believe these substantial revisions directly resolve the clarity and notation issues you raised. We would be sincerely grateful if you could take a moment to review the updated manuscript and share your thoughts.

---

### Official Review · Reviewer_Lqv4 · 2025-11-01

**Soundness:** 3
**Presentation:** 3
**Contribution:** 3
**Rating:** 6
**Confidence:** 2

**Summary:**

The paper considers a meaningful new problem and an effective algorithm. This paper tackles personalized preference inference with extended inductive reasoning. They proposed an algorithm, ALIGNXPLORE, that trains a model that can read a user’s past signals and induce a portable preference description. The profile is then used to condition downstream models to make preference-aligned choices or responses. The induction model is trained through two stages: filtered supervised imitation and RL. Their experiment results show consistent gains over similar-size baselines, competitiveness with larger models, and improved accuracy/efficiency from streaming; the induced profiles generally transfer across downstream models.

**Strengths:**

It is a good paper.
1. The paper reads clearly, and the key ideas, notation, and losses are easy to follow.
2. The idea to use an induction model to generate a readable, explicit, portable personalization profile that can be updated over time is innovative and useful in real applications.
3. The two-stage training, combining supervised imitation and RL, feels direct and complete.
4. The experiments cover a wide range, including single-shot and streaming, cross-model checks, and show strong results.

**Weaknesses:**

There are stil some light weaknesses to consider.
- **LLM-as-Judge reliance.** Results are optimized and evaluated mainly via $R_{\text{jud}}$ (LLM-as-a-judge), with limited human evaluation, so there’s a risk of overfitting to the judge model.
- **Out-of-domain coverage.** P-SOUPS is a solid OOD set, but broader tests (e.g., HelpSteer2, UltraFeedback, SHP, and a persona-style corpus) would strengthen the results.
- **Readability/notation.** In §3.1, the overloaded $R$ (downstream model vs. reward terms) hurts readability even if technically correct.

**Questions:**

1. In the algorithm, it is a two-stage training process. In the first stage, the supervised learning is an imitation learning in my understanding. If the training is good enough, should it conceptually already be able to reach the same level of "golden preference" alone on AlignXtest? In other words, is it fair to compare a supervised trained model with the untrained baselines? We can see the baselines perform much better on P-SOUPS than AlignXtest.

2. In Figure 2, why do the performances not improve with increasing behavior signals after 4 samples? Intuitively, the performances should be at least not worse than before.

3. In Figure 4, if we focus on the original data without smoothness, it looks like the reinforcement itself can reach the same level of performance without a cold start. If so, is the cold-start imitation training really useful?

---

> ### Author Response · Authors · 2025-11-21
> **Response to Reviewer Lqv4 (Part 1)**
>
> > W1: LLM-as-Judge reliance. Results are optimized and evaluated mainly via $R_{jud}$ (LLM-as-a-judge), with limited human evaluation, so there’s a risk of overfitting to the judge model.
>
> We agree that mitigating the risk of overfitting to a single LLM-as-a-judge is essential. For this reason, we designed a comprehensive set of experiments to **rigorously test** our model's generalization capabilities.
>
> 1. **Cross-Model Generalization (Section 4.3, Table 3):** We evaluated our model—trained with a `Qwen2.5-7B-Instruct` judge—against two independent, stronger judges (`QwQ-32B` and `DeepSeek-R1-671B`). As shown in Table 3, our model not only maintains high performance but often **improves** when evaluated by these unseen judges (achieving **68.53** and **67.59** vs. 65.33). This strong cross-model transferability is compelling evidence that our method learns a generalizable skill, not the biases of one specific judge.
>
> 2. **Task-Oriented Validation (Section 4.2, Table 2):** Furthermore, our online evaluation provides task-oriented validation. Here, **GPT-4, conditioned on the ground-truth user preference,** evaluates the final personalized responses. Our model's high win rate in this setup demonstrates that its inferred preferences are not just superficially optimized, but are **more functionally accurate and useful** for the downstream task.
>
> Taken together, these cross-model and task-oriented GPT-4 evaluations provide robust evidence that our method is not overfitting and has successfully learned a generalizable, high-quality preference inference capability. We have revised the manuscript to more prominently highlight these results and their role in validating our approach.
>
> > W2: Out-of-domain coverage. P-SOUPS is a solid OOD set, but broader tests (e.g., HelpSteer2, UltraFeedback, SHP, and a persona-style corpus) would strengthen the results.
>
> We agree that demonstrating generalizability on diverse, out-of-domain (OOD) datasets is crucial for validating the robustness of our method. Following this advice, we conducted a new experiment on a subset of **UltraFeedback**, one of the challenging benchmarks recommended.
>
> **Experimental Design on UltraFeedback:**
>
> To create a rigorous test of personalized preference inference, we constructed preference pairs where the `chosen` response was significantly better in **one specific dimension** (e.g., `helpfulness` score +2) but significantly worse in **all other dimensions** (e.g., `instruction_following`, `honesty`, `truthfulness` scores -2). This setup simulates a user with a highly specific, non-obvious preference (e.g., "prioritize helpfulness above all else, even at the cost of honesty"). The model's task is to infer this specific preference from a history of 8 such examples in a streaming setting and correctly predict the chosen response. The following table shows our model's performance ($\text{ACC}_{\text{jud}}$) on this challenging task, compared to strong baselines:
>
> |Model|$\text{ACC}_{\text{jud}}$ on UltraFeedback
> |-|-
> |**AlignXplore-7B**|**56.03%**
> |DeepSeek-R1-Distill-Qwen-7B|29.45%
> |DeepSeek-R1-671B|43.31%
>
> The results indicate that our model outperforms strong baselines significantly, demonstrating its ability to correctly synthesize these preferences. We will add this new experiment and its analysis to the appendix to further strengthen the paper's claims of generalizability.
>
> > W3: Readability/notation. In §3.1, the overloaded $R$ (downstream model vs. reward terms) hurts readability even if technically correct.
>
> To resolve this ambiguity, we will implement a clear and consistent notation scheme throughout the revised manuscript:
>
> 1. The symbol $R$ will be used **exclusively** for Reward functions (e.g., $R_{\text{jud}}$, $R_{\text{gen}}$), aligning with standard RL terminology.
>
> 2. The downstream personalized response model, currently also denoted by $\mathcal{R}$, will be renamed to $\mathcal{S}$ to clearly signify its role as the **generative model**.
>
> We have applied this simple but important change throughout the methodology and results sections in our updated manuscript to ensure the paper is easy to follow.

---

> ### Author Response · Authors · 2025-11-21
> **Response to Reviewer Lqv4 (Part 2)**
>
> > Q1: In the algorithm, it is a two-stage training process. In the first stage, the supervised learning is an imitation learning in my understanding. If the training is good enough, should it conceptually already be able to reach the same level of "golden preference" alone on AlignXtest? In other words, is it fair to compare a supervised trained model with the untrained baselines? We can see the baselines perform much better on P-SOUPS than AlignXtest.
>
> We will address this in two parts: first by clarifying the conceptual goal of our SFT stage, and second by presenting a new experiment with a fairly-trained baseline to empirically validate our claims.
>
> **1. The Conceptual Role of SFT: Imitation, Not Optimization**
>
> We agree with the framing of SFT as imitation learning. Its objective is not to directly maximize the final reward, but to imitate the teacher model's complex process of generating reasoning and a final preference description. It is well-established that SFT alone struggles to perfect such multi-step tasks. Therefore, its primary role is to provide a strong initialization. The subsequent RL stage is then essential to move beyond imitation and directly optimize for the final task objective, which is critical for reaching optimal performance.
>
> **2. Empirical Validation with a Fairly-Trained Baseline**
>
> To empirically validate this conceptual distinction and ensure a fair comparison, we have conducted a new experiment. We fine-tuned `Qwen2.5-7B-Instruct` using the exact same supervised fine-tuning (SFT) data as our `AlignXplore-7B` model. This creates a direct "SFT-only" baseline that has been exposed to the same in-domain task format. The performance comparison is as follows:
>
> |Model|$\text{AlignX}_{\text{test}}$ (In-Domain)|P-Soups (Informativeness / Style / Expertise)
> |-|-|-
> |Qwen2.5-7B-SFT|61.00%|51.00% / 59.00% / 63.50%
> |**AlignXplore-7B (base)**|65.33%|54.32% / 69.67% / 63.83%
> |**AlignXplore-7B (streaming)** |**71.47%**|**61.30% / 83.00% / 71.33%**
>
> These results lead to two clear conclusions:
> - **Our RL Stage Provides Significant Value:** The new Qwen2.5-7B-SFT baseline achieves a respectable 61.00%, proving SFT effectively teaches the task. However, our full AlignXplore-7B model significantly surpasses this by **+4.33 points (base)** and a remarkable **+10.47 points (streaming)**. This demonstrates our performance gain is not just from SFT, but critically from our reinforcement learning stage.
> - **Our Method Shows Superior Generalization:** This advantage is even more pronounced on the out-of-domain P-Soups benchmark. Our model consistently outperforms the fairly-trained SFT baseline, confirming that our method learns a more generalizable preference inference skill.
>
> In summary, while SFT provides a solid foundation, it is our complete SFT+RL methodology that delivers a significant and, most importantly, generalizable performance improvement. We will add this experiment to the paper to make this point explicit.
>
> > Q2: In Figure 2, why do the performances not improve with increasing behavior signals after 4 samples? Intuitively, the performances should be at least not worse than before.
>
> This counter-intuitive performance drop is a critical finding that highlights two related factors:
>
> 1. **Exceeding Trained Context:** Our `base` model was trained on sequences of 4 examples. Inputs longer than this are out-of-distribution, which degrades performance.
>
> 2. **General Long-Context Degradation:** More generally, with an average signal length of ~385 tokens, providing more than 4 examples creates extremely long prompts (e.g., >3,000 tokens for 8 samples). At these lengths, the context becomes saturated with noise and redundant information, and the model's ability to effectively synthesize key signals diminishes. Both our model and the baseline suffer from this effect, showing it is a general problem.
>
> This reveals a crucial trade-off for non-streaming models: fewer than 4 examples provide insufficient preference signal, while more than 4 introduce overwhelming noise and exceed the trained context length. Consequently, **4 examples represent the optimal balance** for this approach.
>
> This very challenge is the central motivation for our **`streaming`** setting. As shown in Figure 4, the `streaming` model is specifically designed to circumvent this long-context limitation by processing signals iteratively. This results in stable performance as the number of signals grows, demonstrating its superior robustness. We have clarified this analysis in Section 4.4 of our updated manuscript.

---

> ### Author Response · Authors · 2025-11-21
> **Response to Reviewer Lqv4 (Part 3)**
>
> > Q3: In Figure 4, if we focus on the original data without smoothness, it looks like the reinforcement itself can reach the same level of performance without a cold start. If so, is the cold-start imitation training really useful?
>
> In our updated manuscript, the original Figure 4 is now Figure 5. We address this issue as follows:
>
> **1. A Closer Look at Figure 5: Training Stability vs. Overfitting on Seen Data.** The key to interpreting Figure 5 is understanding our training protocol: **RL is run for 4 epochs**. This naturally divides the training curve into two distinct phases:
> - **Phase 1 (Steps 0-50, First Epoch on Unseen Data):** In this initial, critical learning phase, the benefit of the cold-start is undeniable. Our full model (blue curve) begins with a significant head start (reward $\approx$ 0.5) and exhibits high stability. In contrast, the `w/o Cold-start` model (orange curve) starts from a near-random state (reward $\approx$ 0.2) and struggles through a highly inefficient and unstable process. This demonstrates the cold-start's role in providing an effective "warm-up."
> - **Phase 2 (Steps 50-200, Subsequent Epochs on Seen Data):** You are correct that the curves appear to converge in this later phase. However, because the model is now training on **previously seen samples**, the reward measured here is not a reliable indicator of generalization. The apparent convergence is more likely a sign of overfitting to the training distribution rather than a true measure of capability on new data.
>
> **2. The Decisive Metric: Final Performance on Unseen Data (Table 1).** This is precisely why we must rely on a **held-out test set** for the definitive evaluation of the model's final capability. As shown in Table 1, the full `AlignXplore-7B` model achieves an $\text{ACC}_{\text{jud}}$ of **65.33**, while the `w/o Cold-start` model plateaus at a significantly lower **62.80**. This irrecoverable **2.53-point gap** on unseen data is the conclusive proof that the stability and efficiency gained from the cold-start stage translate into a quantifiably superior final model.
>
> In summary, Figure 5 demonstrates that the cold-start stage provides crucial *training efficiency on new data*, while Table 1 proves this efficiency leads to *superior final performance on held-out data*. We have revised the analysis in Section 4.2 of our updated manuscript to make this two-part analysis explicit.

---

> ### Author Response · Authors · 2025-11-26
> **Looking forward to your feedback**
>
> Dear Reviewer Lqv4,
>
> We sincerely appreciate your constructive comments. In response, we have added additional results on UltraFeedback, revised the manuscript to improve overall readability, and provided detailed answers to your questions to address your concerns.
>
> We would greatly appreciate it if you could review our responses at your earliest convenience and share any further feedback you may have.

---

### Author Response · Authors · 2025-12-02
**Rebuttal Summary and Key Updates**

Dear Area Chairs,

We sincerely appreciate your additional efforts in this unprecedented situation following the OpenReview information leak. To assist in your final evaluation, we provide a high-level summary of the current reviewer-author discussion status and the concrete improvements made to our manuscript.

**Core Contribution:** Our paper pioneers the "extended thinking" paradigm within the challenging, inductive domain of personalized preference inference. To address the unique difficulty of evolving user histories, we introduce an efficient streaming mechanism, demonstrating that our approach delivers superior performance and efficiency across diverse scenarios.

### **1. Discussions with Reviewers**

- Reviewer wURy (Score raised: 4 $\rightarrow$ 6):

	Engaged constructively with our rebuttal and explicitly confirmed that their concerns were addressed. They responded positively to our new experiments and raised their score to a 6.

- Reviewer Lqv4 (Score: 6):

	Provided a positive initial assessment ("good paper," "innovative and useful"). While they have not yet responded to the rebuttal, we have addressed their specific questions.

- Reviewers 8Vhm (Score: 4) & VE99 (Score: 4):

	Have not yet replied, but we have addressed all concerns raised in their initial review. We kindly ask that you consider our rebuttal comments in the reviewers’ absence.

### **2. Major Concerns & Improvements in Revised Version**

We have uploaded a revised manuscript that incorporates significant changes based on reviewer feedback:

- Enhanced Clarity and Presentation (Reviewer Lqv4: W3; Reviewer 8Vhm; Reviewer VE99: W1, W2, Q1; Reviewer wURy: W2):

	To thoroughly clarify our methodology and narrative, we overhauled Sections 1, 3, and 4. Crucially, we split the original dense Figure 1 into Figure 1 (Task Overview) and Figure 2 (Training Pipeline) to clearly distinguish the inference task from the training process.

- Strengthened Experimental Validation:

	1. OOD Generalization (Reviewer Lqv4: W2): We added experiments on **UltraFeedback**, proving our method’s strong generalization ability to out-of-domain data.
	2. Baseline Comparison Rigor (Reviewer Lqv4: Q1): We introduced a **Qwen2.5-7B-SFT baseline**, empirically demonstrating the critical necessity of our Reinforcement Learning stage compared to SFT alone.
	3. Human Verification (Reviewer VE99: W3): We conducted **human evaluations** on the quality of generated preference descriptions, confirming our method is reliable, consistent, and stable.
	4. Statistical Significance (Reviewer wURy: W1): We provided **standard deviations** for baselines and model variants across multiple runs, confirming the statistical significance of our improvements.

- Detailed Analysis & Explanations:

	We integrated specific clarifications requested by reviewers into the revised text:
	1. Overfitting risks & Training dynamics (Reviewer Lqv4 W1 & Q2 & Q3: Section 4)
	2. Context length impacts (Reviewer VE99 Q2: Section 4.4)
	3. Ablation insights (Reviewer wURy Q1 & Q2: Section 4.2)

### **3. Summary**

We appreciate the recognition of the meaningful and innovative problem setting (Lqv4, 8Vhm, VE99, wURy), the interpretable and effective methodology (Lqv4, 8Vhm, VE99), and the systematic, extensive empirical validation (Lqv4, VE99, wURy). We are pleased to have resolved the concerns of Reviewer wURy, reaching a consensus that resulted in a raised score. Furthermore, we believe our detailed responses and the revised manuscript effectively address the constructive feedback from Reviewers Lqv4, 8Vhm, and VE99 regarding presentation clarity and additional validation.

We hope this summary facilitates an efficient review of our submission. We are confident that the revised manuscript stands as a rigorous and impactful contribution to the field.

Sincerely,

The Authors of Paper 16822

---

### Meta-Review · Area_Chair_c8dh · 2026-01-02

**Summary:**

Reviewers found the problem meaningful and the results promising, but key concerns remained about whether the paper meets the acceptance bar. The main issue was limited perceived novelty: several reviewers viewed the method as a standard synthetic SFT + GRPO pipeline (close to distillation/composition), and felt the “extended inductive reasoning” framing was not convincingly substantiated beyond longer chains. Reviewers also raised validation risks, including heavy reliance on LLM-as-judge signals and initially insufficient uncertainty/ablation evidence to clearly attribute gains. Finally, clarity and positioning were criticized as hard to follow, with dense figures/notation and an overstated narrative. These concerns motivated my reject recommendation.

**Reviewer Concerns:**

The rebuttal likely addressed the most actionable concerns: it added mean±std over multiple seeds for headline results and key ablations (and clarified the prior t-test), introduced a fair SFT-only baseline to show RL adds value beyond imitation, explained the non-streaming drop via train–test mismatch/long-context saturation (motivating streaming), and added human/variance checks to reduce worries about unstable, open-ended preference descriptions. It also fixed at least one concrete notation/readability issue (overloaded symbols).

Still outstanding or only partially resolved are the bigger-picture critiques: whether the work is truly novel beyond a standard synthetic SFT + GRPO/distillation recipe (and whether “extended reasoning” is more than longer chains), plus limited OOD breadth and residual LLM-as-judge dependence. Some presentation issues remain unverified because the reviewer most concerned about clarity (8Vhm) didn’t confirm improvements, and error bars/qualitative-figure readability weren’t extended to all experiments/figures.

**Reviewer Scores:**

- wURy (4 → 5): This already reflects full participation. They mainly wanted proper error bars and clearer ablations. The authors added multi-seed mean±std (including for ablations) and clarified the reporting, and the reviewer said their concerns were addressed and raised the score.

 - Lqv4 (6 → 6, maybe 7): Their remaining concerns were mild (judge reliance, more OOD tests, and questions about SFT vs RL and why performance saturates after ~4 signals). The rebuttal added an SFT-only baseline, added an UltraFeedback OOD experiment, explained the long-context/mismatch issue, and cleaned up notation. I’d expect them to stay at 6, with a possible bump if the revised paper is clearly easier to read and the new evidence is persuasive.

 - 8Vhm (4 → 5): They were mostly unhappy with readability: a dense figure, confusing notation, and the “inductive reasoning” framing feeling overdone. The authors say they split the figure and rewrote/restructured the method section to be step-by-step. If that rewrite is genuinely clear, I’d expect a +1 to 5, but probably not more.

 - VE99 (4 → 4): Their main issue was novelty: it looks like basic reasoning chains plus a standard synthetic SFT+GRPO distillation recipe, with base performance close to the teacher. The rebuttal helps on side issues (mismatch, human eval, stability), but it may not change that core novelty view. So I’d keep them at 4, with only a small chance of 5 if the revised version makes the non-distillation gains very clear.

---

### Decision · Program_Chairs · 2026-01-26

Reject